# Understanding Conformal Factuality for RAG-based LLMs: Novel Metrics and Systematic Insights

## Abstract

Large language models (LLMs) are powerful generative models which can often generate responses that are plausible but not grounded in factual reality usually referred to as "hallucinations". This is a challenge for using LLMs in applications that require answers that are factually correct. Two approaches have emerged as promising ways to mitigate this issue in the literature: (i) Conformal factuality filtering framework that provides statistical guarantee on the factual accuracy of claims in the final output, but cannot mitigate the hallucinations in the response generation and (ii) retrieval-augmented generation (RAG), which utilizes trusted knowledge bases as reference to guide the generation of response with the aim of reducing hallucinations but does not offer statistical guarantees. In this work, we unite these two approaches by integrating a conformal factuality framework with RAG and systematically study their performance to understand their strengths and limitations. We investigate the role of different key components: reference for generation and scoring functions, sensitivity to calibration data, LLM model capacity, reasoning and robustness to distractors. We propose three new metrics: *non-empty rate*, *non-vacuous empirical factuality*, and *sufficient correctness* to address limitations of standard factuality measures that fail to meaningfully capture usefulness of the output. Our experiments are comprehensive spanning three datasets (FActScore, MATH, and Natural Questions) and multiple model families and sizes. Our results show the importance of designing scoring functions and highlight trade-offs between correctness and informativeness that standard metrics fail to capture. Together, our findings provide insights that are practically useful and sheds light on the importance of re-thinking LLM factuality.

## 1 Introduction

Large language models (LLMs) have demonstrated remarkable capabilities across open-domain question answering, reasoning, and scientific discovery (Brown et al., 2020; Guo et al., 2025; Zhang et al., 2025). Yet, a persistent barrier to their reliable deployment is the phenomenon of *hallucinations*: outputs that are fluent and confident but factually incorrect (Ji et al., 2023; Nadeau et al., 2024; Huang et al., 2025). Such errors are not merely cosmetic. In safety-critical settings, such as medicine, law, or finance, a single fabricated claim can erode trust, propagate misinformation, and incur high societal or financial costs. This makes hallucination mitigation one of the central challenges in advancing trustworthy LLMs.

A rich body of work has emerged to address this challenge with two main directions: (1) retrieval-augmented generation (RAG) and (2) conformal methods. RAG aims to prevent hallucinations by steering the generation process by grounding responses in trusted external knowledge sources. This is usually done by conditioning generation on retrieved passages (Lewis et al., 2020; Gao et al., 2023; Siriwardhana et al., 2023). While RAG reduces the likelihood of unsupported claims, it does not offer statistical guarantees on the factuality of the final response. Conformal prediction (CP) framework, on the other hand, aims to provide a statistical guarantee on the final output, which is often achieved via post-processing of the initial LLM response. CP frameworks usually first decompose the initial LLM output into atomic claims, score each with a given factuality scoring function, and filter those falling below a threshold determined using a calibration data (Mohri & Hashimoto,

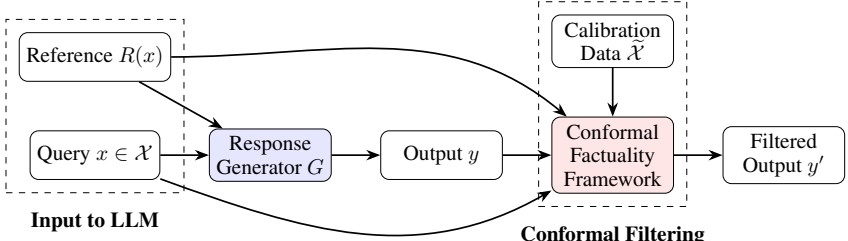

Figure 1: Overview of our framework. Given a query $x$ and retrieved references $R(x)$, the Response Generator $G$ produces an output $y$. The conformal factuality framework utilizes a separate calibration data to determine a threshold used to filter out information from the output $y$ and yield $y'$.

2024; Cherian et al., 2024). This procedure provides formal coverage guarantees but often at the expense of informativeness, since aggressive filtering may yield empty or vacuous outputs. Furthermore, CP filtering cannot improve the usefulness or the accuracy of the LLM response, but only potentially remove hallucinations from the response.

Despite complementary strengths, these two paradigms have primarily been studied in isolation. It is natural to wonder whether combining them allows for significantly improving LLM factuality. There are a few recent works that combine RAG with CP methods (Li et al., 2023; Rouzrokh et al., 2024; Feng et al., 2025), but they are limited in their systematic analysis. A comprehensive understanding of the strengths and limitations of combining RAG and CP methods requires not only integrating the two frameworks but also developing principled ways to evaluate the performance. Standard metrics such as empirical factuality fail to capture the usefulness of the final answer, e.g., an empty answer is by default factually correct.

**Our Contributions**: We *systematically* investigate the integration of conformal factuality filtering with RAG-based LLMs to establish a unified framework for hallucination mitigation. Our comprehensive experiments span diverse datasets, model families, and scoring functions. Furthermore, we evaluate robustness against distribution shifts and distractors, shedding light on both the capabilities and limitations of this approach.

Beyond empirical analysis, we also address the limitations of current evaluation practices. To this end, we introduce three complementary metrics: *non-empty rate* and *non-vacuous empirical factuality*, which jointly capture both correctness and information retention, and *sufficient correctness*, which measures whether an output contains enough correct information to infer the final answer to the query. These novel metrics capture the trade-off between factual correctness and informativeness, providing practical insights and tools for future work on hallucination mitigation.

## 2 PROBLEM SETTING, DATASETS, MODELS, AND METRICS

Let $x \in \mathcal{X}$ denote an input query. Let $R(x)$ denote the reference material sufficient to answer $x$. We assume the existence of an oracle retriever that can retrieve $R(x)$ such that the true answer $y^\star$ is either in or can be deduced from $R(x)$. A response generator $G$, instantiated as a large language model (LLM), is prompted with both $x$ and $R(x)$ to produce an output $y = G(x, R(x))$. Even with reference, LLMs can produce hallucinations in the generated response (Huang et al., 2025). The goal of the overall system is to create a final output $y'$ such that each statement in $y'$ is factually correct at a user's expected level, e.g., 85%, while being useful in answering the query. Conformal filtering methods provide such statistical guarantees, which we describe below.

**Conformal Filtering.** Let $\widetilde{\mathcal{X}}$ denote a calibration set disjoint from $\mathcal{X}$. For each query $x \in \widetilde{\mathcal{X}}$, we obtain a set of claims $\{c_i\}$ by parsing the corresponding $y$ using a parser $P$. These claims are then scored by a factuality scoring function $f$. Given a user-specified error level $\alpha \in (0, 1)$, the conformal filtering framework determines a threshold $\tau_\alpha$ from the calibration set. Specifically, $\tau_\alpha$ is chosen as the $\frac{\lceil (n+1)(1-\alpha) \rceil}{n}$ quantile of a set of candidate thresholds, where $n = |\widetilde{\mathcal{X}}|$. For each calibration query $x$, the candidate threshold is defined as the smallest value of $\tau$ such that all claims surviving

above $\tau$ are factual: $\inf\{\tau \mid \forall c \in F(\tau),\ c \text{ is factual}\}$, where $F(\tau) = \{c \mid f(c) > \tau\}$ denotes the set of claims scoring above $\tau$. The resulting guarantee is that $\mathbb{P}\big(\forall c \in F(\tau_\alpha),\ c \text{ is factual}\big)\ \geq\ 1 - \alpha$. At inference time, each generated output $y$ is decomposed into atomic claims by a parser $P$, yielding $C(y) = \{c_i\}_{i=1}^k$. Each claim $c_i$ is scored with $f$, and those exceeding the threshold are retained: $C'(y) = \{\, c_i \mid f(c_i) > \tau_\alpha,\ i = 1, \ldots, k \,\}$. Finally, the retained claims $C'(y)$ are merged by a merger $M$ into a single filtered response: $y' = M(C'(y))$. Figure A.1.3 in the Appendix illustrates the whole pipeline of the conformal filtering system.

## 2.1 Scoring Functions

Scoring functions $f$ are the core component of the conformal factuality framework, as they determine whether a claim is retained or filtered. We study two main families of scoring functions:

**(a) Entailment-based scorers** are natural language inference (NLI) models to assess whether the reference text supports a claim. We use both document-level and sentence-level variants. For **document-level entailment**, the entailment score is computed directly between the entire reference $R(x)$ and the claim $c_i$. We use an entailment model (Laurer et al., 2022) trained on the DocNLI dataset (Yin et al., 2021) for this.

In the sentence-level setting, entailment scores are computed between the claim $c_i$ and each sentence in $R(x)$ and then aggregated in two different ways. The first is **conservative entailment**, where a claim is marked as contradictory if any sentence in the reference contradicts it. It is marked as entailed only if there are no contradictions and there is at least one sentence supporting it. The second is **average entailment**, which averages the scores of all non-neutral sentence-level comparisons. For sentence-level entailment, we use `roberta-large-mnli` as the entailment model (Liu et al., 2019).

**(b) LLM-based scorers.** In this family, we prompt a language model[1] to assign a factuality score to each claim. We explore the design space by varying five dimensions: (i) the inclusion of retrieved references $R(x)$, (ii) evidence highlighting within $R(x)$, (iii) the use of Chain-of-Thought (CoT) reasoning, (iv) output granularity (continuous $[0, 1]$ vs. Boolean), and (v) evaluation consistency (single generation vs. averaging over five independent generations).

## 2.2 Datasets

We perform evaluations on three datasets that span open-ended summarization, mathematical reasoning, and question answering tasks. This diversity allows us to assess both the factual reliability and the task-level utility of our approach.

- **FActScore** dataset (Min et al., 2023) consists of 601 individuals, each paired with a Wikipedia page. Queries are: "Tell me a paragraph biography about `[person]`," where the reference $R(x)$ is the Wikipedia page of the person. Because no canonical ground-truth answers are provided, this dataset is well-suited for evaluating factuality in *open-ended generation* and for testing whether models can produce faithful summaries grounded in external references. We additionally consider **FActScore Rare**, a subset of 198 queries focusing on less well-known individuals. This subset probes the robustness of models when the model's parametric knowledge is not enough to answer the question, a regime where hallucinations are more likely, if a reference is not given.
- **MATH** dataset (Hendrycks et al., 2021) contains 12,446 competition-style mathematics problems spanning five difficulty levels and seven categories. Each problem provides a question $x$ and a ground-truth answer $y^\star$. To construct reference materials, we prompted `gpt-5-nano` to generate prerequisite knowledge relevant to solving each problem, which serves as $R(x)$ [2].
- **Natural Questions (NQ)** dataset (Kwiatkowski et al., 2019) consists of 10K real-world queries collected from search engines. Each query is annotated with both a long answer and a short answer; we use the long answer as the reference $R(x)$ and the short answer as the ground-truth.

Together, these datasets provide coverage over distinct capabilities: factual summarization (FActScore), mathematical reasoning (MATH), and reference-based question answering (NQ).

---

[1] Prompts provided in Appendix A.5.8.

[2] Prompts provided in Appendix A.5.14

## 2.3 LANGUAGE MODELS

We evaluate our framework across both open-source and proprietary language models in order to systematically evaluate parts and whole of the Factuality and RAG pipeline under varying architectures, reasoning modes, and parameter scales.

**Open-source models.** Our open-source suite includes multiple families. The `Qwen3` models (Yang et al., 2025) (0.6B, 4B, and 8B parameters) are evaluated both in their base form and in a reasoning-enabled variant, `Qwen3-Think`, where reasoning with the `<think></think>` tag is enabled. This contrast allows us to study whether reasoning-oriented training improves factuality scoring and filtering. To broaden architectural diversity, we also include `Llama-3.2-Instruct` (Dubey et al., 2024) (1B and 3B parameters), `SmolLM2-Instruct` (Allal et al., 2025) (135M, 360M, and 1.7B parameters), and `gpt-oss-20b` [3] (Agarwal et al., 2025) (21B parameters with 3.6B active).

**Proprietary model.** In addition to open-source models, we employ `gpt-5-nano` (gpt, 2025), a proprietary model, primarily as an auxiliary component. We use it for tasks such as claim parsing, factuality labeling, and claim merging, where higher accuracy in controlled subtasks helps construct reliable evaluation pipelines. Importantly, the proprietary model is not the subject of our evaluation, but rather supports the analysis of the open-source models.

The chosen models are designed to probe three orthogonal dimensions: (i) *model architecture*, by comparing across families; (ii) *reasoning capability*, by contrasting `Qwen3` with `Qwen3-Think`; and (iii) *model scale*. This diversity allows us to assess how each factor influences factuality scoring, and to highlight regimes where smaller, more efficient models suffice.

## 2.4 EVALUATION METRICS

We evaluate the performance using both established factuality measures and new metrics designed to capture aspects of informativeness that existing metrics overlook. We use three widely used criteria:

- **Empirical Factuality (EF)** measures the fraction of outputs $y'$ in which all retained claims $C'(y)$ are factual. By convention, an empty claim set $C'(y) = \varnothing$ is treated as factual, which can artificially inflate EF when filtering is overly aggressive. *Higher EF is better*, as it indicates stronger factual reliability.
- **Power** quantifies the average proportion of true claims retained. *Higher Power is better*, since it means fewer correct claims are lost.
- **False Positive Rate (FPR)** measures the fraction of non-factual claims that survive. *Lower FPR is better*, as it reflects stronger suppression of hallucinations.

These metrics only provide limited insight into the overall usefulness of the final answer. Each of the statements in an LLM response could be factually correct while still not being informative enough to answer the input query. Furthermore, a vacuous or empty output is factually correct by definition, but is not useful, and when the reference does contain information to provide a correct answer, an empty answer is an indication of failure. While EF, Power, and FPR capture factuality and error rates, they fail to penalize vacuous but "factual" outputs. To address this, we propose the following *novel* evaluation metrics.

- **Non-empty Rate (NR)**, the fraction of outputs that preserve at least one claim. *Higher NR is better*, rewarding informative responses rather than empty ones.
- **Non-vacuous Empirical Factuality (NvEF)**, which computes EF only over non-empty outputs. *Higher NvEF is better*, reflecting factuality conditional on informativeness.
- **Correctness** measures the fraction of outputs equivalent to the ground-truth answer $y^\star$. *Higher Correctness is better*, though this metric is intentionally strict and more applicable to the dataset with ground truth answers.
- **Sufficient Correctness (SC)**, which holds if the filtered output contains enough correct information, relative to a reference $R(x)$, to infer the correct answer. *Higher SC is better*, as it captures end-task utility. This metric is inspired by sufficient context introduced in (Joren et al., 2025).

NR, NvEF are claim-level, while correctness and sufficient correctness are at the final task-level outcome. Together, these metrics balance factual reliability, informativeness, and task-level utility, ensuring that evaluation reflects not only safety (removing hallucinations) but also usefulness (retaining an adequate signal for the end task).

---

[3]Results are deferred to Appendix A.2.8

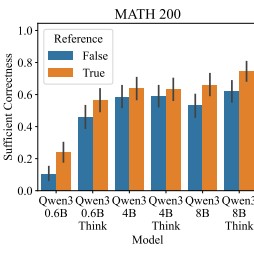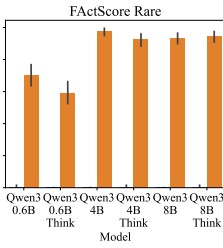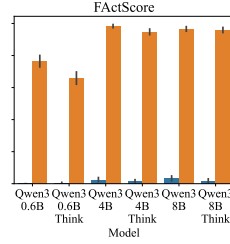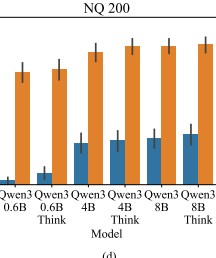

Figure 2: Sufficient correctness (SC) of Qwen3 models (0.6B, 4B, 8B) on four datasets (MATH-200, FActScore-Rare, FActScore, NQ-200), with and without access to references. Across model sizes and datasets, providing references consistently improves generation quality.

We now design a series of experiments to systematically analyze: (i) the role of references, (ii) the design of scoring functions, and (iii) robustness to distributional shifts and adversarial inputs.

## 3 IMPACT OF REFERENCES

We begin by isolating the role of references, asking: *How much does retrieved reference improve generation quality before filtering is even applied?* To study this, we evaluate outputs $y$ produced by the response generator $G$ under two conditions: query-only generation $y = G(x)$ and query-plus-reference generation $y = G(x, R(x))$. We then measure sufficient correctness and compare across FActScore, FActScore-Rare, **MATH-200** (a 200-example subset of MATH), and **NQ-200** (a 200-example subset of Natural Questions). These datasets help provide insights into whether conditioning on $R(x)$ improves different types of tasks: factual summarization, reasoning, and question answering.

**Results.** Figure 2 compares sufficient correctness (SC) with and without references for the Qwen3 family. Note that the FActScore and NQ datasets, reference plays a huge role, which highlights its importance when the model is not able to have all information memorized. On MATH, the gap between the two settings narrows as model size increases, while overall SC improves, suggesting that larger models possess stronger reasoning ability and can better leverage reference material. For Qwen3-0.6B, enabling *think* reasoning produces a large jump in SC, further highlighting the role of reasoning capacity. In contrast, for FActScore and NQ, where math reasoning plays a more minor role, enabling *think* has little effect. Lastly, we observe that there are diminishing returns in terms of increasing the model size in the Qwen3 model family. The improvement in terms of SC when we increase the model size from 4B to 8B is smaller compared to when we improve the model size from 0.6B to 4B. Overall, providing references consistently improves generation quality. We observe similar trends for SmolLM2 and Llama-3.2 (see Figure 12 and 13 in the Appendix), and therefore provide references to the generator in all subsequent experiments.

## 4 SCORING FUNCTIONS

Having shown that reference improves baseline quality, we now turn to conformal filtering, which offers statistical guarantees. Its backbone is the scoring function for factuality. In this section, we examine how prompting strategies, references, model choice, and different scores affect performance.

### 4.1 PROMPTING STRATEGIES FOR LLM SCORERS

We begin by examining the effect of prompting variations for LLM-based scoring functions, including: (i) highlighting supporting evidence, (ii) enabling chain-of-thought reasoning, (iii) scalar vs Boolean scoring, and (iv) consistency averaging. References are always provided to both $f$ and $G$. We run this experiment on **FActScore**, **MATH-1K** (a 1,050-example subset of MATH), and **NQ-1K** (a 1,000-example subset of Natural Questions).

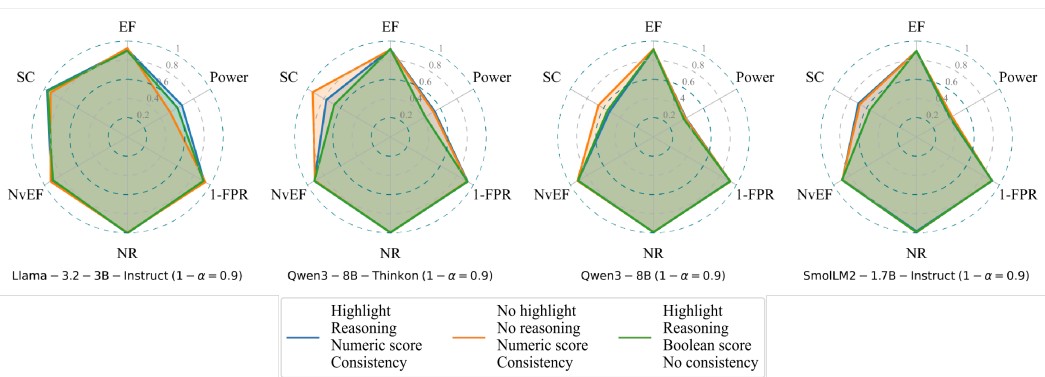

Figure 3: We evaluated 16 prompting strategies across three LLMs, presenting three representative approaches for clarity. Results demonstrate that: (i) prompting models to generate numeric scores consistently outperforms boolean scoring; (ii) sampling multiple responses uniformly improves performance; however, (iii) incorporating chain-of-reasoning or response highlighting as grounding mechanisms does not yield reliable performance gains across models.

**Results.** Figure 3 summarizes the effect of prompting strategies on model confidence scores across three LLM families. While no single strategy emerges as universally optimal, several consistent patterns are observed. First, instructing models to output numeric scores reliably outperforms boolean judgments, suggesting that scalar responses better capture gradations of uncertainty. Second, sampling multiple responses provides consistent gains, indicating that aggregation reduces variance and stabilizes confidence estimates. In contrast, more elaborate mechanisms, such as chain-of-reasoning prompts or response highlighting, fail to deliver consistent improvements and sometimes even degrade performance. Our findings demonstrate that generic prompting heuristics prove inadequate for conformal factuality tasks, necessitating architecture-specific prompting strategies and highlighting the importance of principled, fine-grained optimization approaches such as TextGrad (Yuksekgonul et al., 2024) for future research.

### 4.2 ROLE OF REFERENCES IN SCORING

In this section, we evaluate how providing references to the LLM-based scoring function improves it. Outputs $y = G(x, R(x))$ are generated using `gpt-5-nano`, while open-source models serve as scorers. This is because API calls are faster than using our GPU for inference. When varying reference access, we fix all other prompt settings (CoT enabled, scalar scoring, and consistency averaging). Experiments are conducted on FActScore, MATH-1K, and NQ-1K.

**Results.** Figure 4 shows the performance under various metrics for the model confidence scores with and without reference provided to the scoring function for the MATH-1K dataset using `Qwen3-4B` as the scorer. We observe that when a reference is introduced to the scoring function, the power is consistently higher than in the case where no reference is given to the scoring function. Similarly, for the non-empty rate (NR), the scoring functions with reference have an advantage when the target factuality is of a larger size. These results show the benefit of feeding the reference to the scoring function. Therefore, in the subsequent experiments, we provide a reference to our scoring function.

### 4.3 MODEL CHOICE FOR LLM-BASED SCORERS

We compare different open-source models as scoring functions $f$, using `gpt-5-nano` as the generator $G$. All prompts include references, require highlighting and CoT, produce scalar scores, and apply consistency averaging. This experiment assesses how the scorer model family and scale affect factuality filtering.

**Results.** Our experiments reveal that scaling LLMs does not guarantee improved confidence calibration in conformal factuality, as shown in Figure 5. While the `Llama` series exhibits a consistent gain on model confidence score with larger models, this trend breaks for other families. The `SmolLM2` series shows no systematic benefit, and in the case of `Qwen3`, scaling even degrades

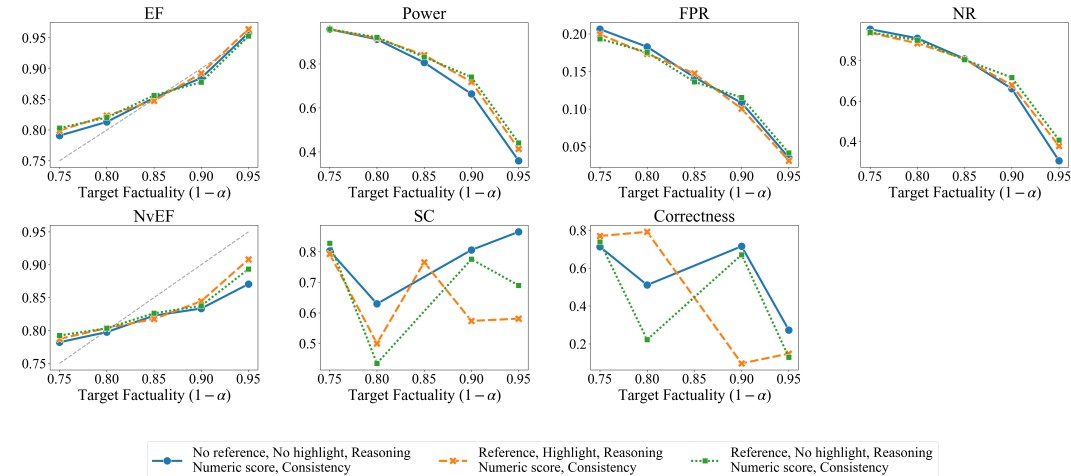

Figure 4: Performance of model confidence score on MATH-1K dataset with and without reference provided to scoring functions using `Qwen3-4B` as the LLM-based scorer.

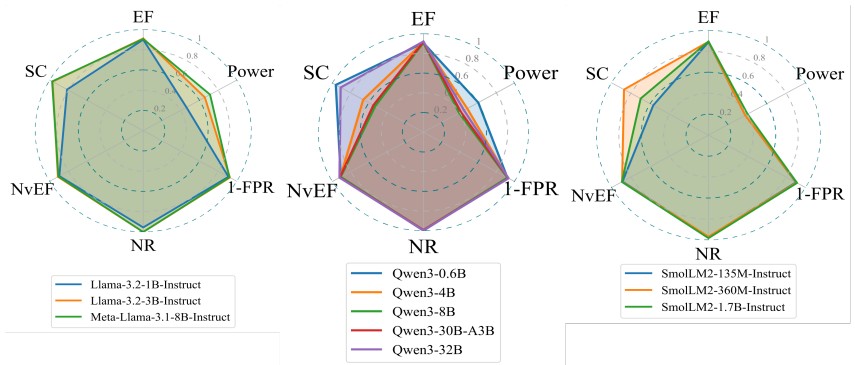

Figure 5: We conducted experiments across three model series of varying scales. Results demonstrate that increasing model size does not consistently improve performance in conformal factuality. While scaling the `Llama` series yields consistent improvements in confidence score accuracy, this pattern does not hold for the `Qwen3` and `SmolLM2` series. Notably, for the `Qwen3` series, scaling may degrade performance.

performance. These heterogeneous scaling behaviors challenge the assumption that model size universally improves calibration quality, suggesting that progress in conformal factuality will require more fine-grained design beyond naive scaling.

## 4.4 DIFFERENT FAMILY OF SCORING FUNCTIONS

Beyond model confidence scores, we also examine entailment-based scoring functions. We compare confidence-based filtering with entailment-based filtering on FActScore, MATH-1K, and NQ-1K datasets to assess whether entailment signals offer additional advantages in factuality filtering.

**Results.** Figure 6 presents a direct comparison between model confidence scores[4] and entailment-based scoring. Notably, entailment-based scores consistently match or exceed model confidence, despite the entailment model being substantially smaller than the target LLMs[5]. Together with our scaling analysis 4.3, these findings challenge the assumption that larger models inherently yield

---

[4]Under the best prompting strategy we identified in Section 4.1 for each LLM.

[5]We employ `DeBERTa` and `RoBERTa` to compute entailment scores, achieving computational efficiency gains of more than two orders of magnitude compared to LLM-based confidence scoring methods. For detailed comparisons of model parameters and computational complexity, see Table 1.

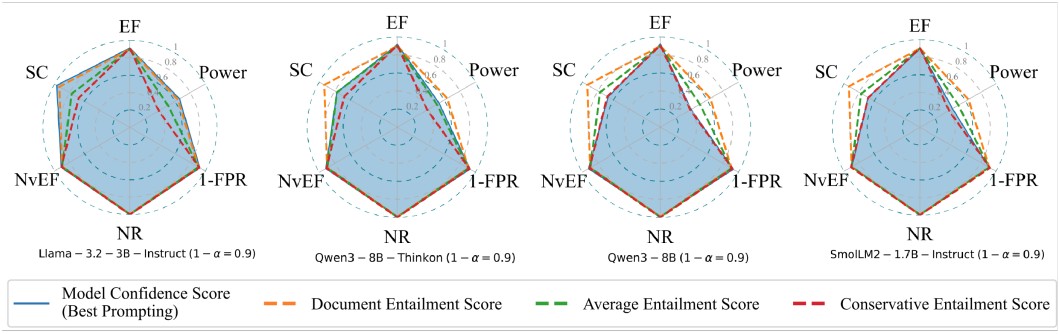

Figure 6: Using the best-performing prompt for each LLM identified in Section 4.1, we compare entailment-based scores against the model confidence score. Across our evaluated settings, the entailment-based scores match or exceed the model-confidence baseline, despite the entailment model being substantially smaller than the target LLMs.

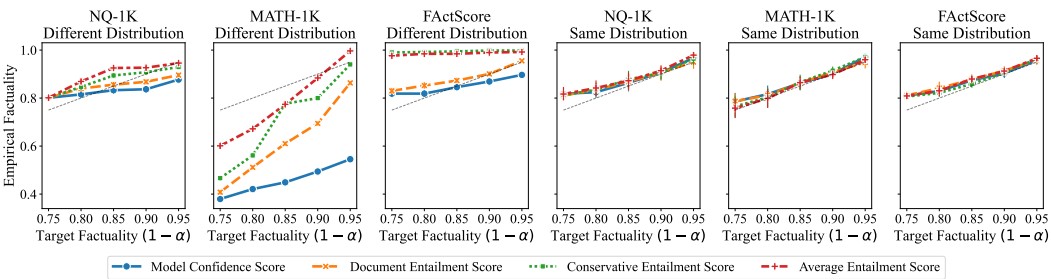

Figure 7: Empirical factuality (EF) on FActScore, MATH-1K, and NQ-1K, under two calibration settings: (i) calibration claims generated in Mohri & Hashimoto (2024), which comes from a different distribution (ii) calibration claims drawn from the same distribution as the test data. We use `Qwen3-4B` as the scoring function. The results show how the distribution shift in the calibration set affects conformal factuality guarantees.

better-calibrated confidence estimates, instead, highlighting that targeted, lightweight verifiers can deliver both computational efficiency and superior performance.

## 5 ROBUSTNESS OF SCORING FUNCTIONS

Although scoring functions differ in their design and accuracy, their real-world usefulness depends on robustness: can they maintain reliability under distribution shifts or adversarial perturbations? We therefore stress-test our framework under challenging conditions.

### 5.1 ROBUSTNESS TO CALIBRATION DISTRIBUTION SHIFT

Conformal factuality assumes calibration and test data are exchangeable. We evaluate robustness when this assumption is violated by using mismatched calibration data. Specifically, we compare calibration on (i) 50 human-annotated claims from Mohri & Hashimoto (2024) (`gpt-4` generated claims, denoted as MH) versus (ii) 50 randomly selected queries from the held-out test half. We evaluate empirical factuality using `Qwen3-4B`, `SmolLM2-360M-Instruct`, and `Llama-3.2-3B-Instruct` across FActScore, MATH-1K, and NQ-1K datasets.

**Results.** Figure 7 compares empirical factuality (EF) under different sources of calibration data, with test outputs generated and scored by `Qwen3-4B`. When calibration data come from a different distribution (generated by `gpt-4`), EF sometimes falls below the target level (grey line) for certain datasets and scoring functions. While the entailment based scorers are robust to this distribution shift, this robustness is not consistent across language models: switching $G$ and $f$ to `SmolLM2-360M-Instruct` or `Llama-3.2-3B-Instruct` yields different behaviors for the

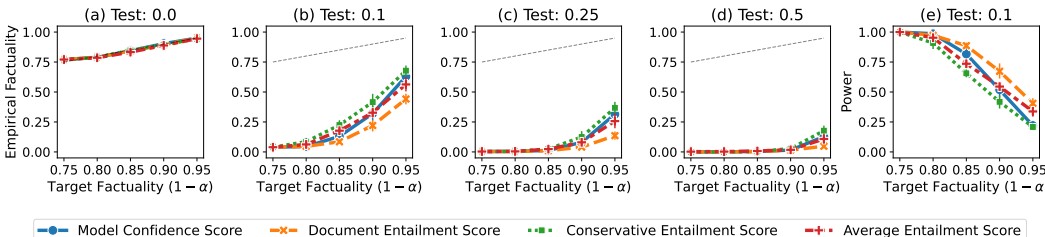

Figure 8: (a) - (d) Empirical factuality versus target factuality (1- $\alpha$) for Qwen3-4B on the FActScore dataset. Each panel corresponds to a different distractor injection rate in the test set (0.0, 0.1, 0.25, 0.5). The gray dashed diagonal represents perfect calibration ($y = x$), where empirical factuality matches the target. (e) Power versus target factuality (1 - $\alpha$) when the injection rate is 0.1.

scoring functions that seem robust in the `Qwen3-4B` setting (see Figures 29 and 28 in the appendix). This shows that the CP filtering is sensitive to the distribution shifts between the claims during test time and the calibration set. Note that this arises even when the LLM used to generate the response is switched for the same dataset. So, the calibration data should be collected using the same LLM that the system is going to be deployed with.

## 5.2 ROBUSTNESS TO DISTRACTORS

In real-world scenarios, the LLM outputs may contain claims that are plausible but incorrect. This may be due to the fact that LLMs themselves are easily distracted by irrelevant information (Shi et al., 2023). To simulate this setting, we replace a proportion of factual claims in each test query with distractors generated by `gpt-5-nano`. We then evaluate whether the conformal factuality framework can reliably distinguish correct content from these plausible but non-factual claims. Experiments are conducted on FActScore, MATH-1K, and NQ-1K datasets. We defer the discussion of how we create the distractors to Appendix A.3.

**Results.** Figure 8 (a)-(d) show that as the distractor rate increases, empirical factuality (EF) drops sharply. This degradation occurs because adding distractors to the test set likely violates the exchangeability assumption underlying the conformal factuality framework, causing EF to fall below the target level. Although EF can increase when the target factuality is set very high, this comes at the cost of a substantial loss in power, as shown in Figure 8 (e). Our results indicate that CP filtering with current scoring functions is not robust to distractors, underscoring the need for improved scoring functions that maintain robustness in their presence.

### 5.2.1 CAN DISTRACTION-AWARE CALIBRATION HELP?

A natural way to mitigate the effect of distractors is to anticipate their presence by using distraction-aware calibration. To study the efficacy of this approach, we extend the previous setting by introducing distractors not only into the test set but also into the calibration set. This models potential distribution shifts caused by distractor content and evaluates whether conformal filtering remains reliable when calibration data include such claims. Since the true level of distractors in practice is unknown, we vary the amount of distractors in calibration and test sets independently. This setup enables us to assess both under-estimation and over-estimation of distractor prevalence. Experiments are conducted on FActScore, MATH-1K, and NQ-1K datasets.

**Results.** In Figure 9, we show the result for the setting with a test set with a distractor proportion of 0.25 and varying the levels of distractors in the calibration set. When the fraction of distractors is underestimated, the EF is still far below the target EF (Figure 9 (a)). From Figure 9 (b)-(c), we can see that introducing a large enough fraction of distractors to the calibration set can bring up the empirical factuality. However, we note that this incurs a high cost on the non-empty rate. As we see in Figure 9 (d)-(e), the non-empty rate drops significantly when distractors are introduced to the calibration set. When both calibration and test contain no distractors, we have a much higher non-empty rate compared to the case where we inject 25% distractors into both the calibration and test sets. This happens likely due to the thresholds found by the CP framework becoming more stringent.

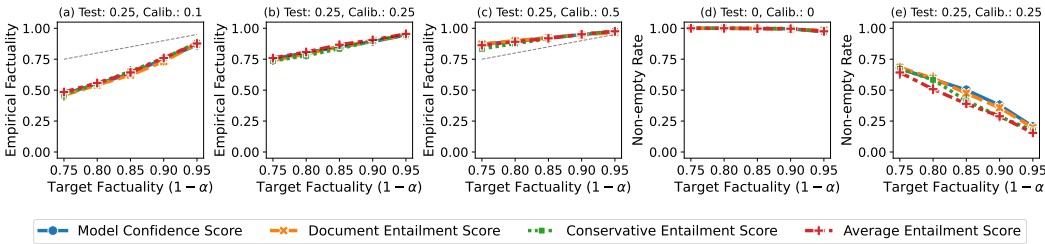

Figure 9: (a) - (c) Empirical factuality versus target factuality $(1-\alpha)$ for Qwen3-4B on the FActScore dataset when the test set is injected with 25% distractors. We vary the proportion of distractors in the calibration set from (0.1, 0.25, 0.5). As the proportion matches, we can see that the empirical factuality rises to the $y = x$ line. (d)-(e) Comparison of non-empty rate when both the test and calibration sets contain no distractors and contain 25% of the distractors. Although introducing distractors to the calibration set can achieve target factuality, the non-empty rate suffers.

Therefore, the scoring functions cannot distinguish the distractors that are factually incorrect from the factually correct claims.

## 6 RELATED WORKS

Many studies show that LLMs are prone to hallucination Nadeau et al. (2024); Huang et al. (2025) despite their impressive capabilities in summarization, dialogue, and coding Achiam et al. (2023); Zhang et al. (2024); Nam et al. (2024).

**Retrieval-augmented generation (RAG)** improves LLM performance on knowledge-intensive tasks by grounding responses in retrieved external context Lewis et al. (2020); Joren et al. (2025); Gao et al. (2023). While RAG improves the LLM generation, the output can still contain hallucinations Huang et al. (2025). Moreover, LLM may not utilize the reference well, especially that information in the middle, a phenomenon known as lost-in-the-middle Liu et al. (2023); Ravaut et al. (2023); Chen et al. (2023); Tang et al. (2023), which may be caused by the use of rotary positional embedding (RoPE) Su et al. (2024); Huang et al. (2025). Another reason for this to happen is that in the pretraining data, the most salient information is located at the beginning or the end, rather than in the middle Ravaut et al. (2023); Huang et al. (2025).

**Conformal prediction (CP)** to filter non-factual content from LLM outputs has emerged as a promising strategy Vovk et al. (2005); Angelopoulos & Bates (2021); Mohri & Hashimoto (2024); Cherian et al. (2024). These methods not only improve factuality but also offer a statistical guarantee: $\mathbb{P}(\text{Output is factual}) \geq 1 - \alpha$. For instance, Mohri & Hashimoto (2024) introduced *conformal factuality*, which scores claims in the output and removes those below a threshold.

## 7 CONCLUSION

We systematically investigated the role of key components in hallucination mitigation, including the use of references for generation and scoring, sensitivity to calibration data, model capacity, reasoning ability, and robustness to distractors. To address limitations of standard factuality measures, we proposed three new metrics: *non-empty rate*, *non-vacuous empirical factuality*, and *sufficient correctness*, which better capture the usefulness of filtered outputs. Our experiments span three diverse datasets (FActScore, MATH, and Natural Questions) and three model families. Our results highlight the importance of scoring function design and reveal trade-offs between correctness and informativeness that standard metrics overlook. Overall, our findings provide practical insights and underscore the need to rethink how LLM factuality is measured and enforced.

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

# A APPENDIX

## A.1 EXTENDED RELATED WORKS

### A.1.1 RETRIEVAL-AUGMENTED GENERATION

Retrieval-augmented generation (RAG) improves LLM performance on knowledge-intensive tasks by grounding responses in retrieved external context Lewis et al. (2020); Joren et al. (2025); Gao et al. (2023). Formally, given a query $x$, a retriever $R$ returns context $R(x)$ that supplements the model's parametric knowledge, guiding generation toward more factually correct outputs. In this work, we assume access to an oracle retriever that always provides relevant, accurate references. While RAG is powerful, it does not provide statistical guarantees on the factuality of its outputs.

### A.1.2 CONFORMAL PREDICTION

Despite their impressive capabilities in summarization, dialogue, and coding Achiam et al. (2023); Zhang et al. (2024); Nam et al. (2024), LLMs are prone to hallucination Nadeau et al. (2024); Huang et al. (2025). One promising mitigation strategy applies conformal prediction to filter non-factual content from LLM outputs Vovk et al. (2005); Angelopoulos & Bates (2021); Mohri & Hashimoto (2024); Cherian et al. (2024). These methods not only improve factuality but also offer a statistical guarantee: $\mathbb{P}(\text{Output is factual}) \geq 1 - \alpha$. For instance, Mohri & Hashimoto (2024) introduced *conformal factuality*, which scores claims in the output and removes those below a threshold calibrated on held-out data. The choice of scoring function $f$ is therefore critical to the effectiveness of the framework.

### A.1.3 CONFORMAL PREDICTION AND RAG

Recent work has begun integrating conformal prediction into retrieval-augmented generation (RAG) to provide statistical reliability guarantees. TRAQ Li et al. (2023) applies conformal prediction at both the retriever and generator stages, ensuring with high probability that a semantically correct answer is included in the output set. Conformal-RAG Feng et al. (2025) instead operates at the sub-claim level, filtering unreliable statements to guarantee factuality across domains. Conflare Rouzrokh et al. (2024) focuses on the retrieval stage, calibrating similarity thresholds so that retrieved contexts contain the true answer with user-specified confidence. While these approaches provide important coverage guarantees at different stages of the pipeline, they largely assess correctness in isolation. In contrast, our work unifies retrieval and filtering and introduces new metrics—non-empty rate, non-vacuous empirical factuality, and sufficient correctness—that explicitly capture the trade-off between correctness and informativeness, which existing frameworks do not address.

## A.2 EXTENDED RESULTS

### A.2.1 IMPACT OF REFERENCES

Section 3 in the main paper studies the impact of references on the initial responses generated by the LLM. There, we look at how sufficient correctness changes while we vary whether to give a reference to the LLM or not. Figure 11 illustrates the sufficient correctness of models from the `Qwen3` family, as well as two frontier models: `gemini-2.5-pro` (Team et al., 2023) and `gpt-5.1` (gpt, 2025) on the FActScore Rare dataset. Although the frontier models perform better when a reference is given, we can see that `Qwen3-4B` is comparable to these frontier models when the reference text is given. Figure 12 and Figure 13 show the same comparison for the models from the `Llama3` family and `SmolLM2` family, respectively. We observe that across model sizes and datasets, providing references consistently improves generation quality.

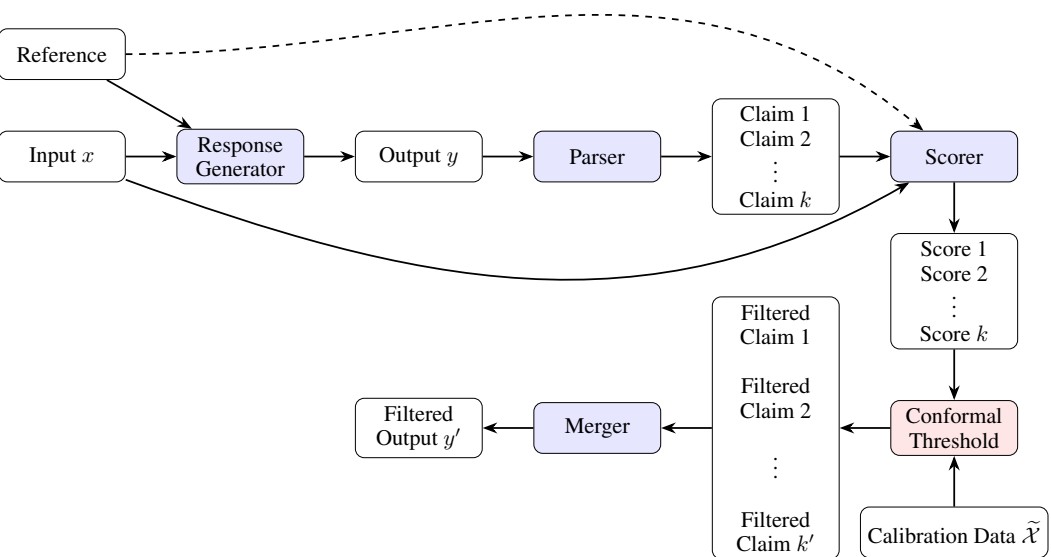

Figure 10: Given an input $x$ and a reference text related to $x$, the Response Generator produces an output $y$, which is then parsed by the Parser into a list of claims. Each individual claim is subsequently scored by the Scorer, conditioned on the input $x$ and, optionally, the reference text. These scores are passed to the conformal prediction algorithm, which filters out claims whose scores fall below a learned threshold. Finally, the remaining claims are merged into a single paragraph and returned to the user.

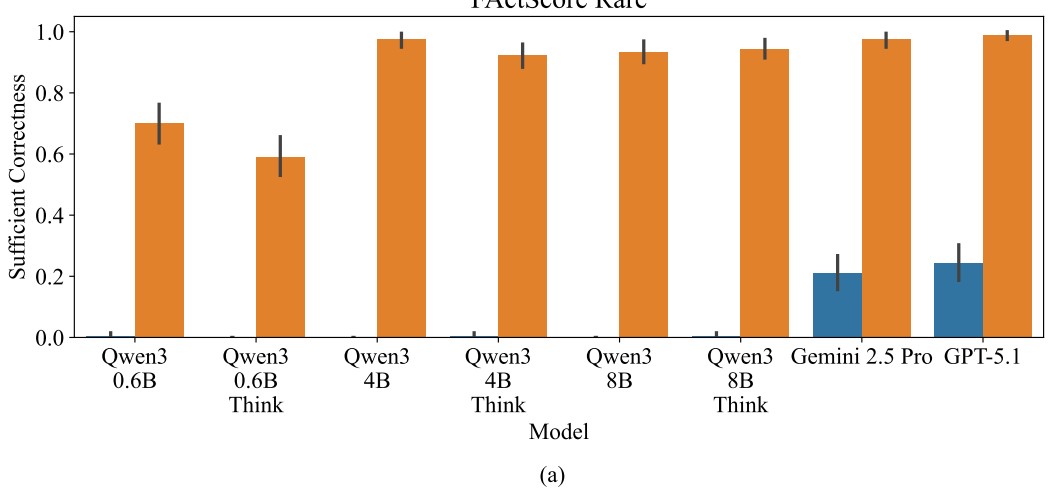

(a)

Figure 11: Sufficient correctness (SC) of `Qwen3` and some frontier models on FActScore-Rare dataset, with and without access to references. Across these models, providing references consistently improves generation quality.

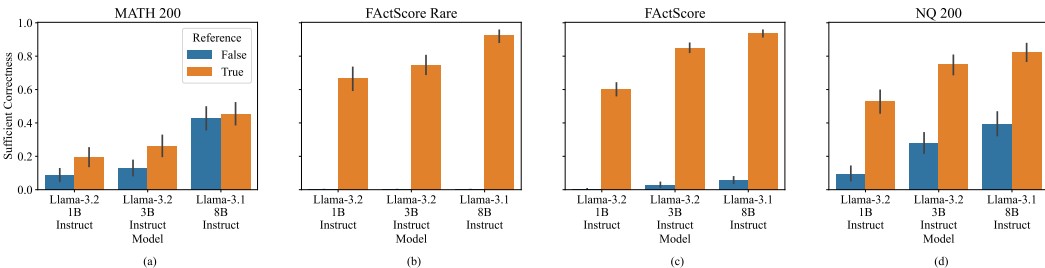

Figure 12: Sufficient correctness (SC) of `Llama 3.1` and `Llama 3.2` models on four datasets (MATH-200, FActScore-Rare, FActScore, NQ-200), with and without access to references. Across model sizes and datasets, providing references consistently improves generation quality.

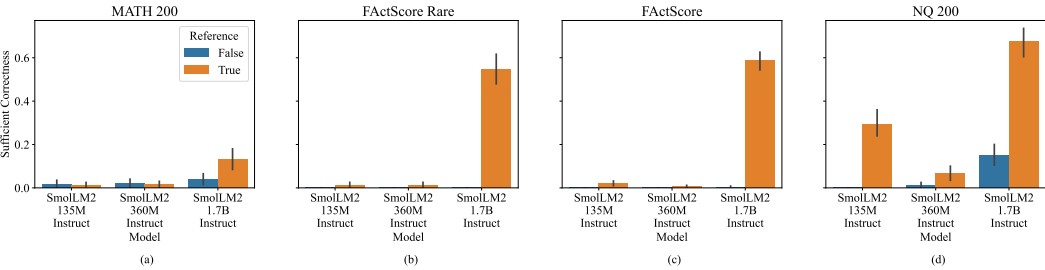

Figure 13: Sufficient correctness (SC) of `SmolLM2` models on four datasets (MATH-200, FActScore-Rare, FActScore, NQ-200), with and without access to references. Across model sizes and datasets, providing references consistently improves generation quality.

### A.2.2 PROMPTING STRATEGIES FOR SCORERS

Figure 14 and Figure 15 show the overall performance comparison of model confidence score across different prompting strategies with different target factuality on the FActScore dataset and MATH-1K dataset, respectively.

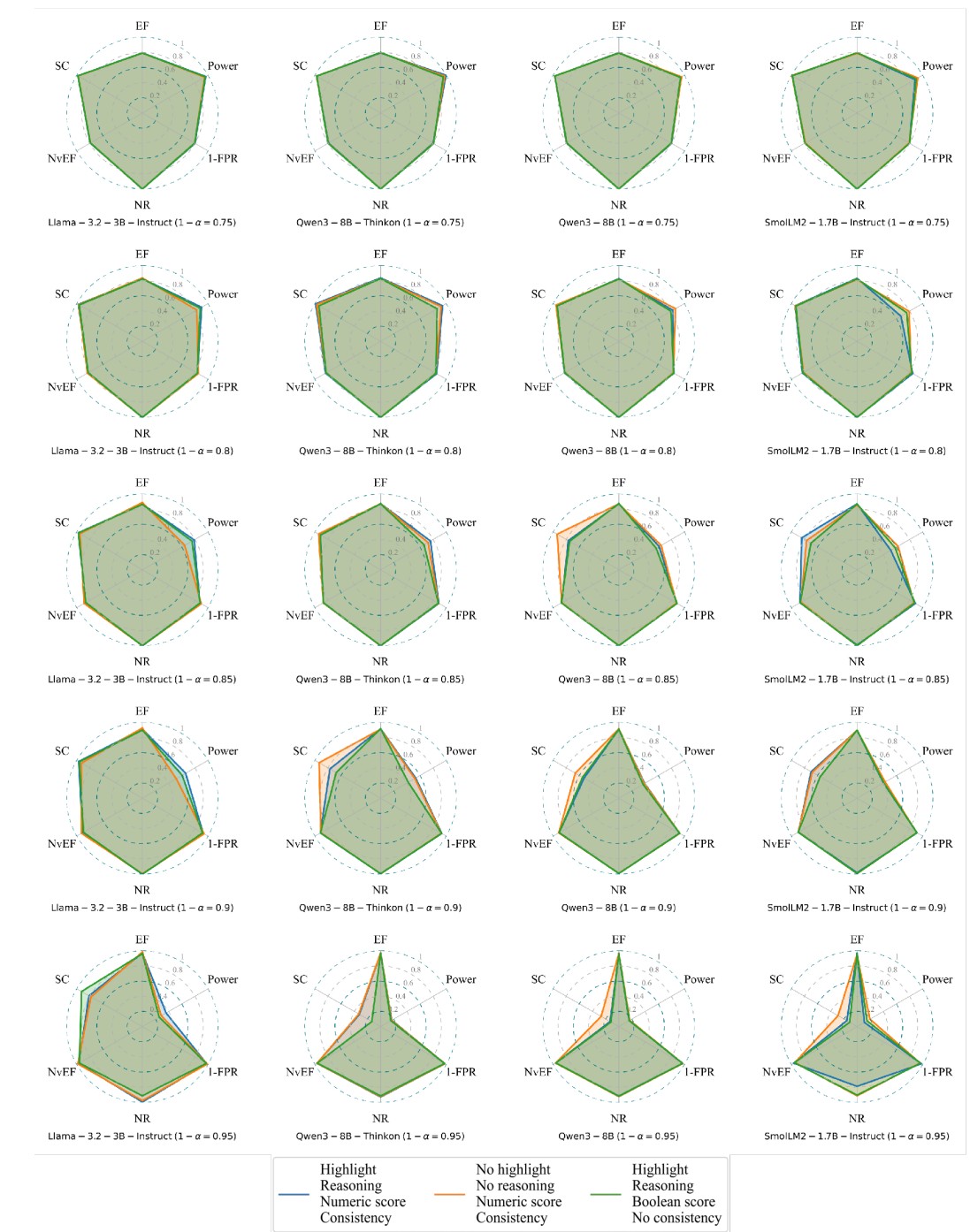

Figure 14: Overall performance comparison of model confidence scores across different prompting strategies with different target factuality $(1 - \alpha)$ on FActScore dataset.

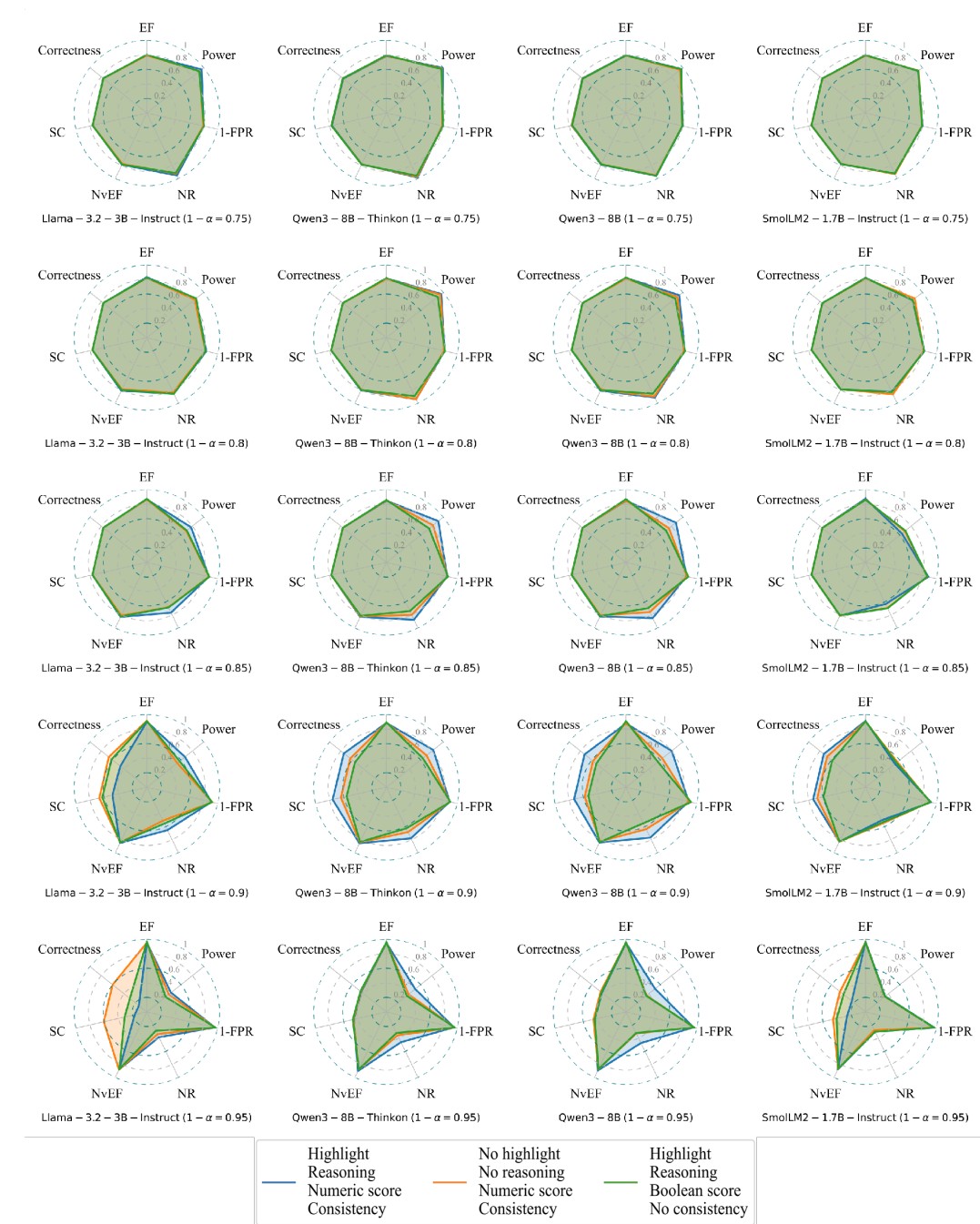

Figure 15: Overall performance comparison of model confidence scores across different prompting strategies with different target factuality $(1 - \alpha)$ on MATH-1K dataset.

### A.2.3 ROLE OF REFERENCES IN SCORING

To investigate the role of references to the model confidence score, we compare the case of giving and not giving the reference text to the scoring function under various metrics. Figure 16, Figure 4, and Figure 17 show the performance of various metrics using `Qwen3-4B` on the FActScore, MATH-1K, and NQ-1K datasets, respectively. Similarly, Figure 18, Figure 19, and Figure 20 show the performance of various metrics using `Qwen3-8B` on the FActScore, MATH-1K, and NQ-1K datasets.

Figure 21-23 and Figure 24-26 show the performances on the three datasets using `Llama-3.2B` and `SmolLM2-1.7B`, respectively.

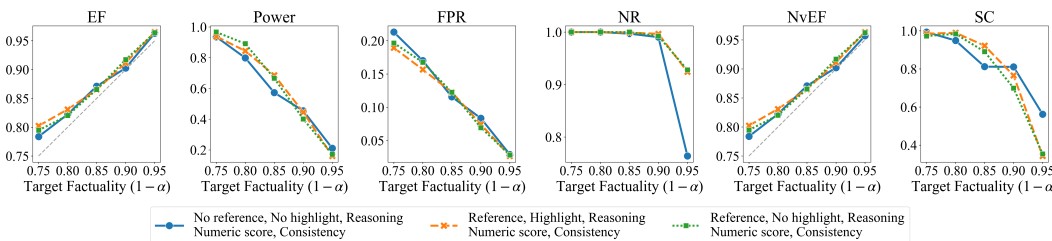

Figure 16: Performance of model confidence score on FActScore with and without reference provided to scoring functions using `gpt-5-nano` and `Qwen3-4B`.

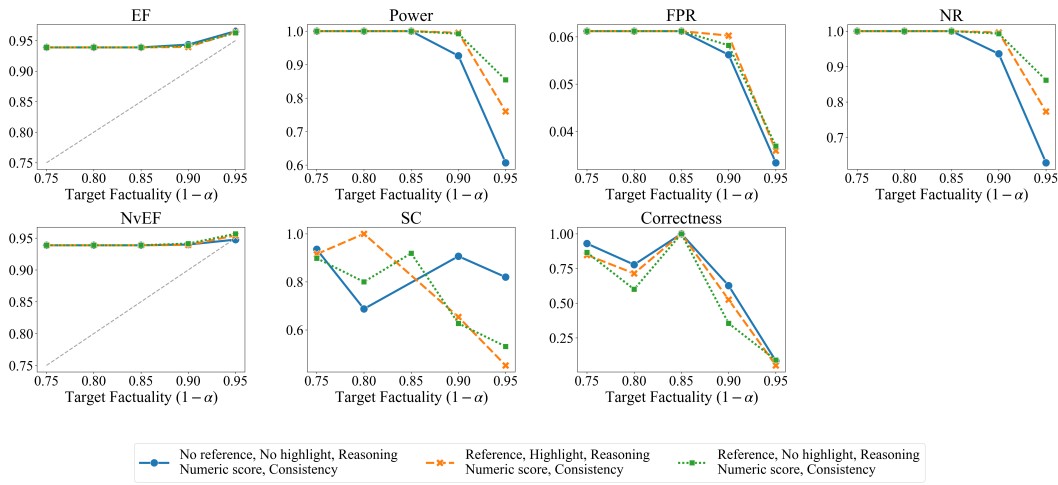

Figure 17: Performance of model confidence score on NQ-1K dataset with and without reference provided to scoring functions using `gpt-5-nano` and `Qwen3-4B`.

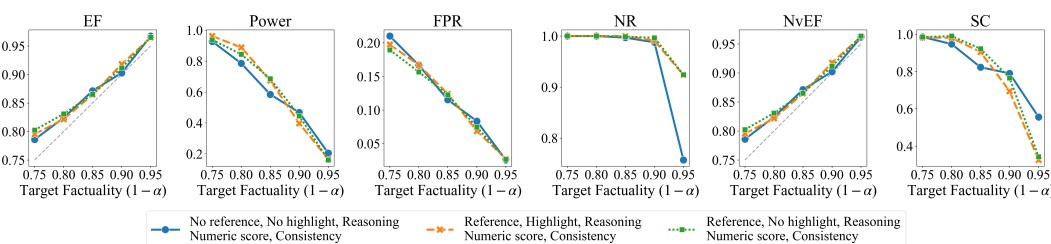

Figure 18: Performance of model confidence score on FActScore dataset with and without reference provided to scoring functions using `gpt-5-nano` and `Qwen3-8B`.

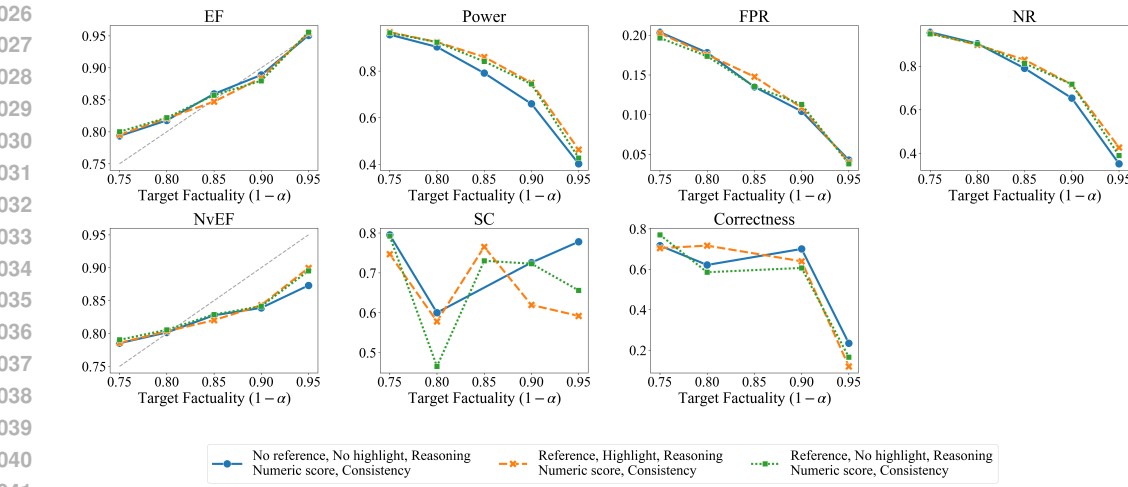

Figure 19: Performance of model confidence score on MATH-1K dataset with and without reference provided to scoring functions using `gpt-5-nano` and `Qwen3-8B`.

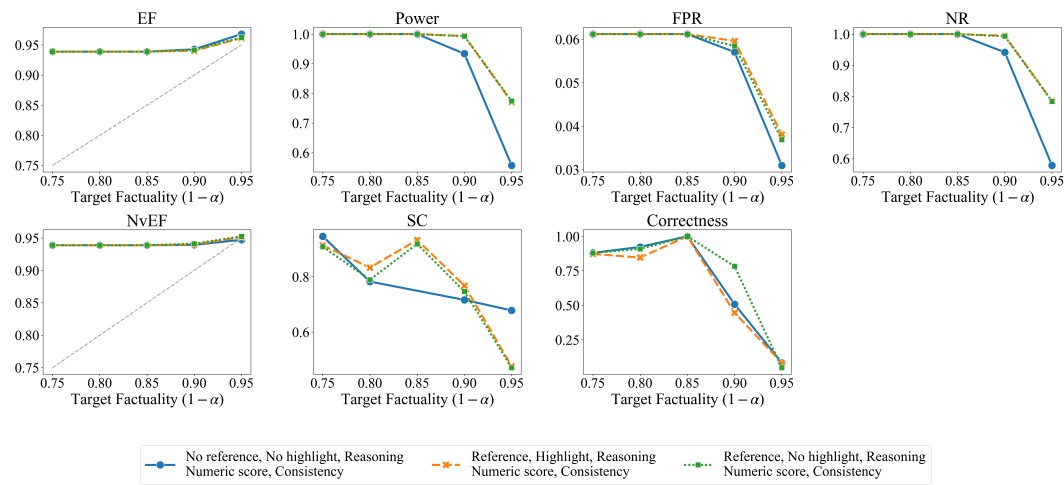

Figure 20: Performance of model confidence score on NQ-1K dataset with and without reference provided to scoring functions using `gpt-5-nano` and `Qwen3-8B`.

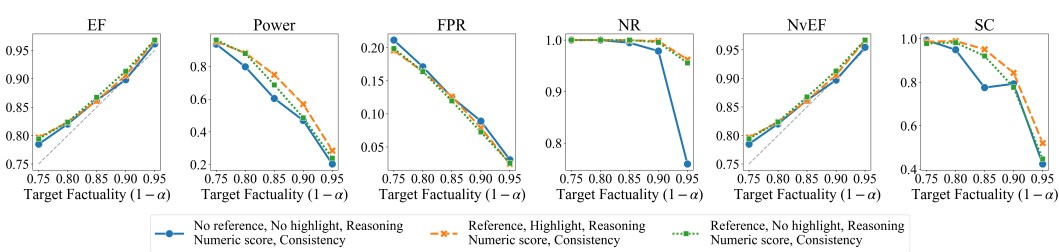

Figure 21: Performance of model confidence score on FActScore dataset with and without reference provided to scoring functions using `gpt-5-nano` and `Llama-3.2-3B`.

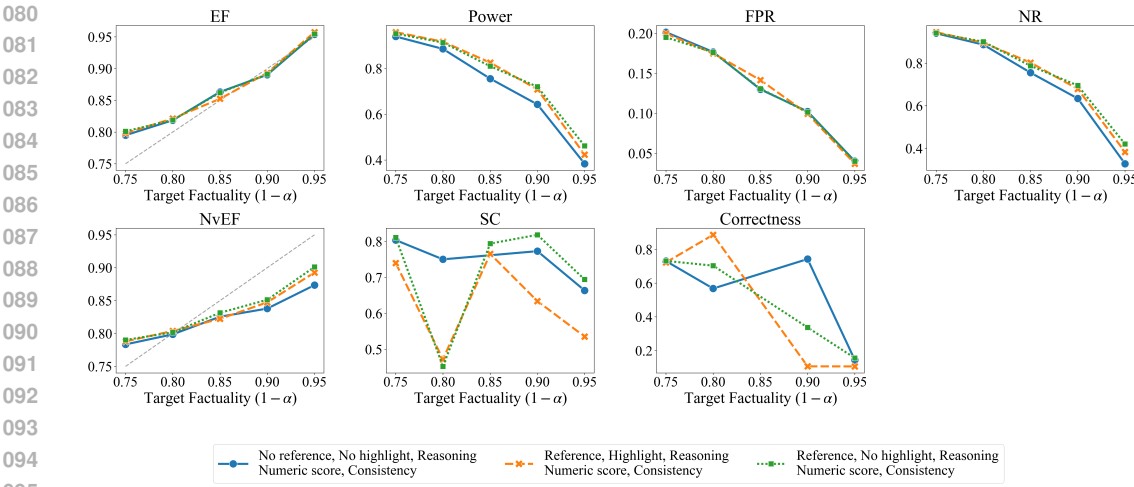

Figure 22: Performance of model confidence score on MATH-1K dataset with and without reference provided to scoring functions using `gpt-5-nano` and `Llama-3.2-3B`.

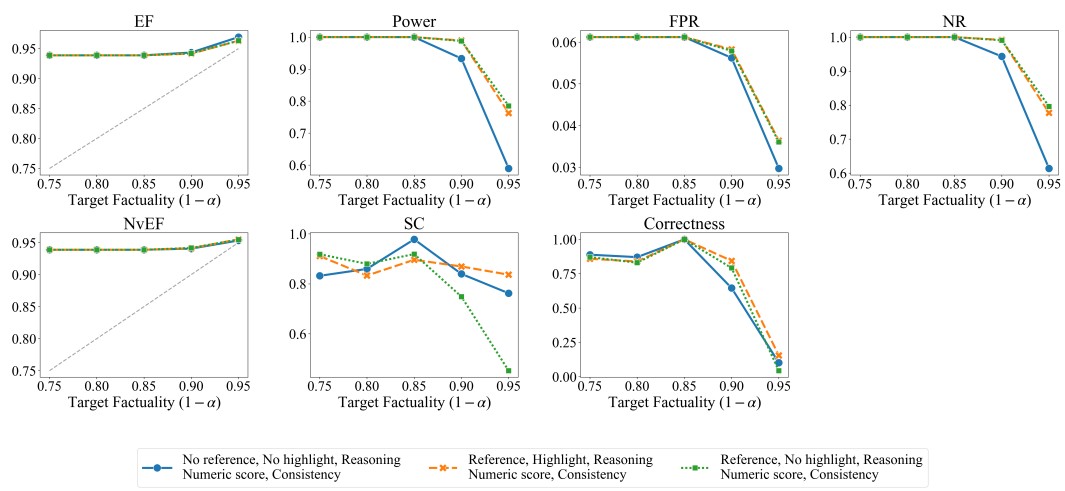

Figure 23: Performance of model confidence score on NQ-1K dataset with and without reference provided to scoring functions using `gpt-5-nano` and `Llama-3.2-3B`.

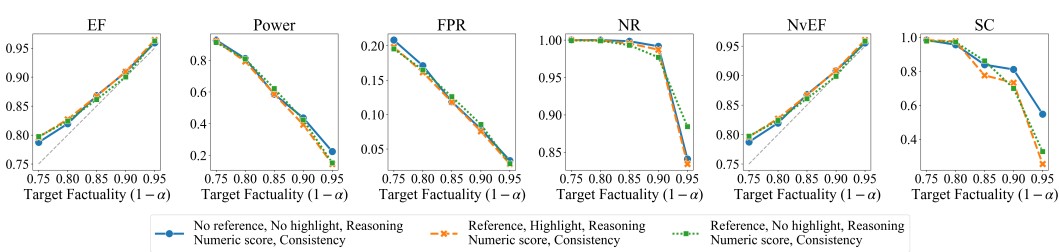

Figure 24: Performance of model confidence score on FActScore dataset with and without reference provided to scoring functions using `gpt-5-nano` and `SmolLM2-1.7B`.

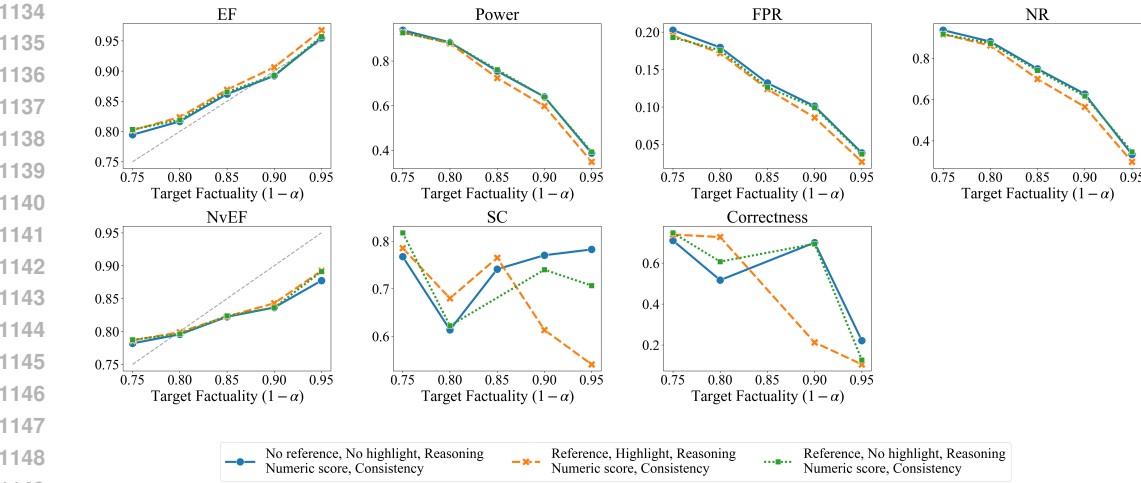

Figure 25: Performance of model confidence score on MATH-1K dataset with and without reference provided to scoring functions using `gpt-5-nano` and `SmolLM2-1.7B`.

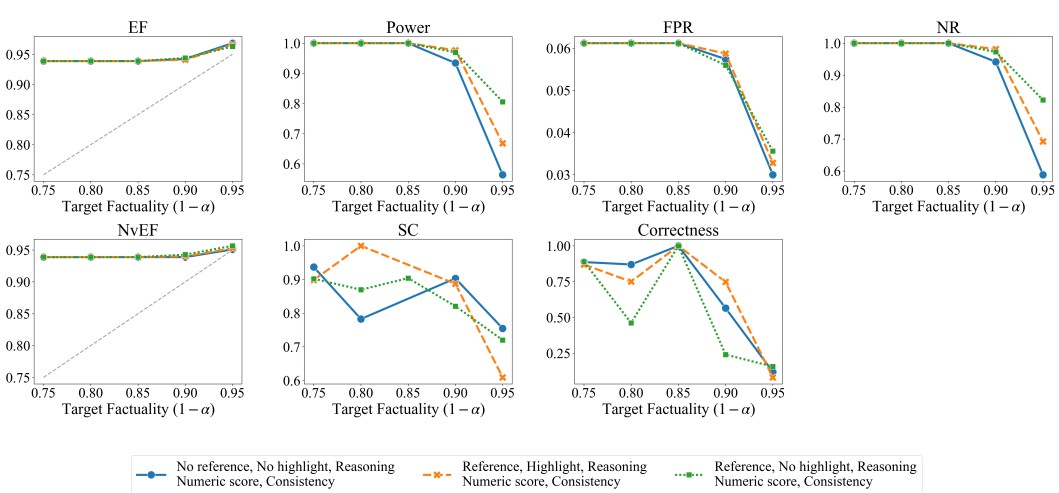

Figure 26: Performance of model confidence score on NQ-1K dataset with and without reference provided to scoring functions using `gpt-5-nano` and `SmolLM2-1.7B`.

### A.2.4 MODEL CHOICE FOR SCORERS

We compare different open-source models as scoring functions $f$, using `gpt-5-nano` as the generator $G$. All prompts include references, require highlighting and CoT, produce scalar scores, and apply consistency averaging. Figure 27 shows the assessment of how scorer model family and their scale affect factuality filtering with different target factuality $1 - \alpha$.

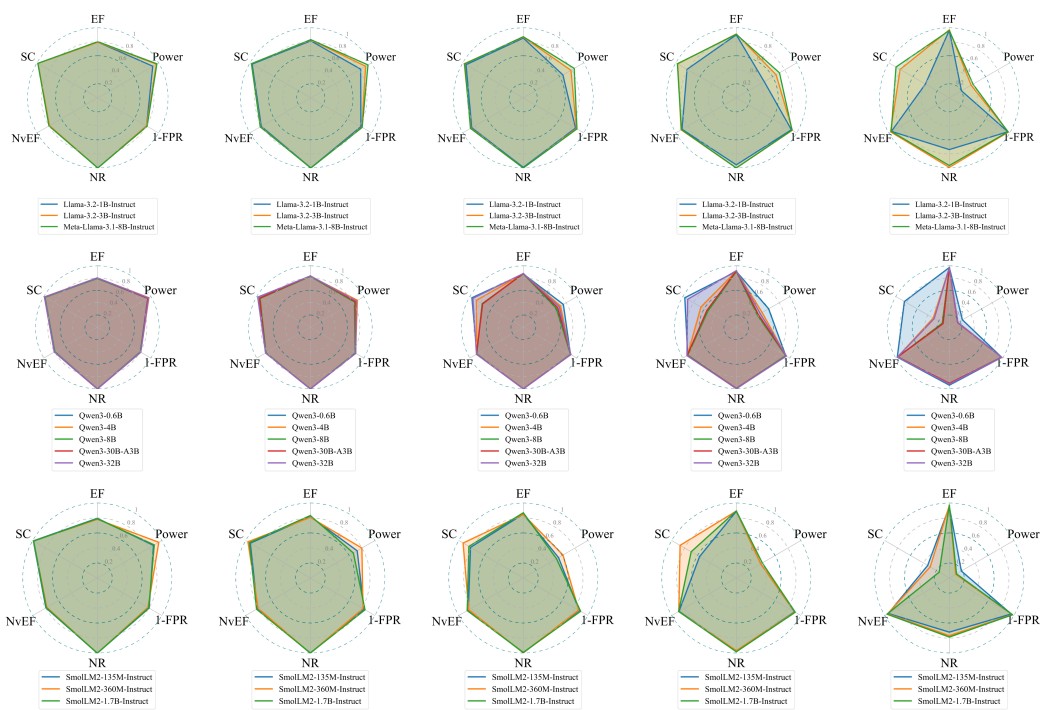

Figure 27: Overall performance comparison of model confidence scores across different model scales with different target factuality $1 - \alpha$.

### A.2.5 ROBUSTNESS TO CALIBRATION DISTRIBUTION SHIFT

Although scoring functions differ in their design and accuracy, their real-world usefulness depends on robustness: can they maintain reliability under distribution shifts or adversarial perturbations? Figure 28 and 29 show that for the 3 datasets, when there is a distribution shift, the factuality guarantee no longer holds.

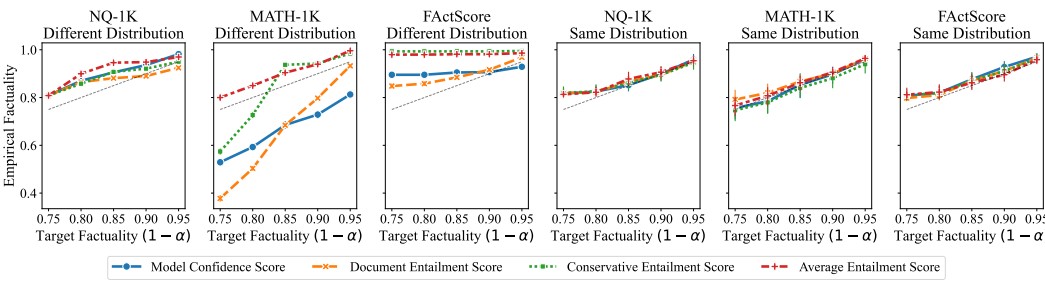

Figure 28: Empirical factuality (EF) on FActScore, MATH-1K, and NQ-1K, under two calibration settings: (i) calibration claims generated in Mohri & Hashimoto (2024), which comes from a different distribution (ii) calibration claims drawn from the same distribution as the test data. We use `Llama3.2-3B` as the scoring function. The results show how distribution shift in the calibration set affects conformal factuality guarantees.

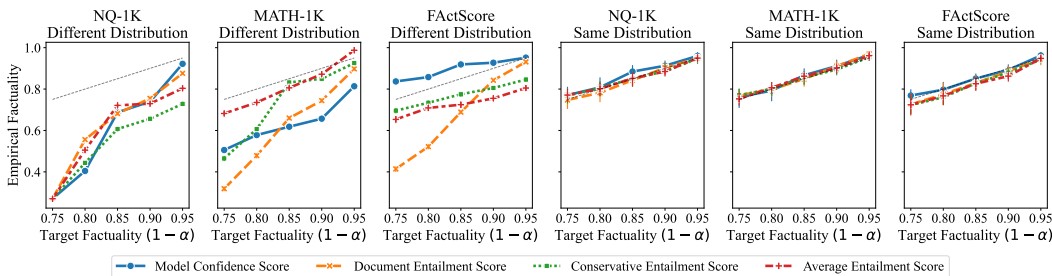

Figure 29: Empirical factuality (EF) on FActScore, MATH-1K, and NQ-1K, under two calibration settings: (i) calibration claims generated in Mohri & Hashimoto (2024), which comes from a different distribution (ii) calibration claims drawn from the same distribution as the test data. We use `SmolLM2-360M` as the scoring function. The results show how distribution shift in the calibration set affects conformal factuality guarantees.

### A.2.6 ROBUSTNESS TO ADVERSARIAL DISTRACTORS

Figure 30 and Figure 31 show that for `Llama3.2-3B-Instruct` and `SmolLM2-1.7B-Instruct`, as the distractor rate increases, empirical factuality drops sharply on the FActScore dataset. Similar observations are also found on the MATH-1K and NQ-1K dataset, as shown in Figure 32-37 on MATH-1K and NQ-1K dataset.

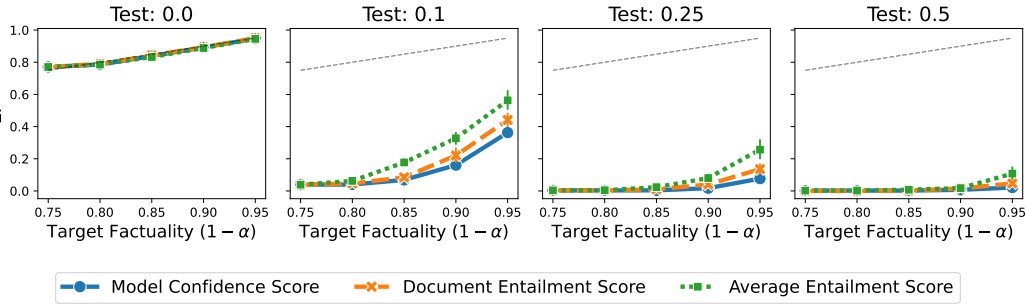

Figure 30: Empirical factuality under varying test hallucination proportions (0.0 to 0.5) with fixed clean calibration data. Three scoring methods compared on FActScore dataset with `Llama-3.2-3B-Instruct`.

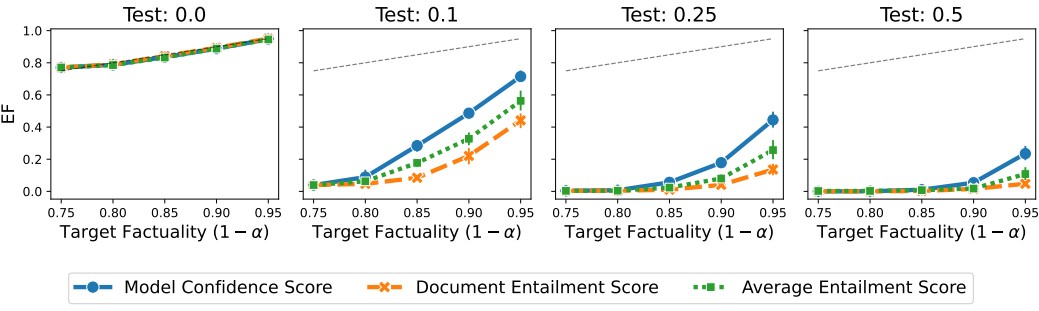

Figure 31: Empirical factuality under varying test hallucination proportions (0.0 to 0.5) with fixed clean calibration data. Three scoring methods compared on FActScore dataset with `SmolLM2-1.7B-Instruct`.

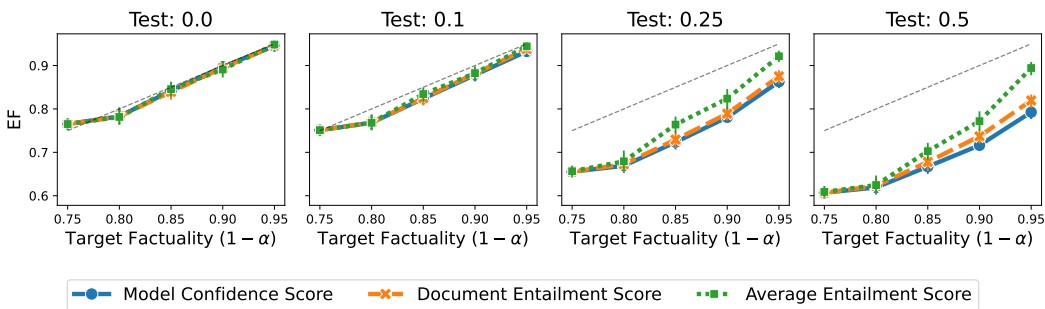

Figure 32: Empirical factuality under varying test hallucination proportions (0.0 to 0.5) with fixed clean calibration data. Three scoring methods compared on MATH-1K dataset with `Qwen3-4B`.

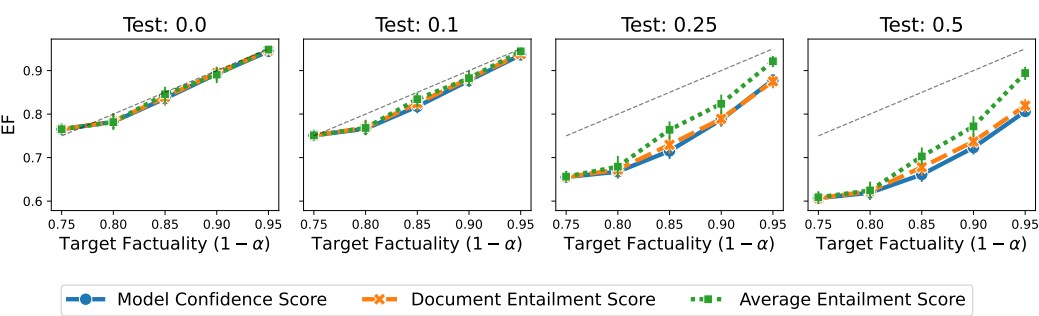

Figure 33: Empirical factuality under varying test hallucination proportions (0.0 to 0.5) with fixed clean calibration data. Three scoring methods compared on MATH-1K dataset with `Llama-3.2-3B-Instruct`.

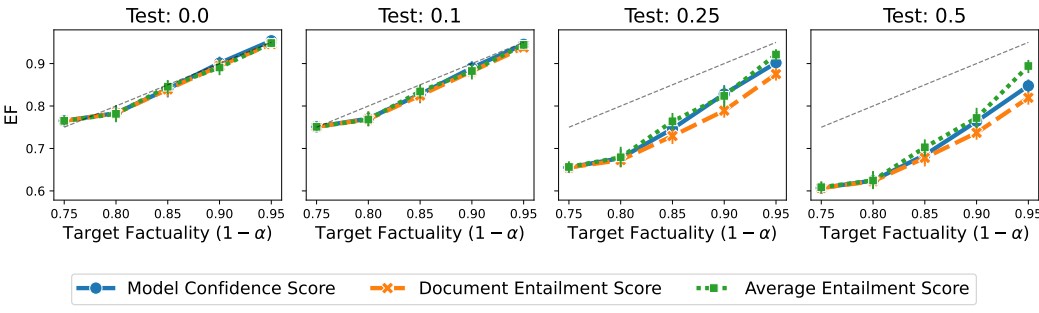

Figure 34: Empirical factuality under varying test hallucination proportions (0.0 to 0.5) with fixed clean calibration data. Three scoring methods compared on MATH-1K dataset with `SmolLM2-1.7B-Instruct`.

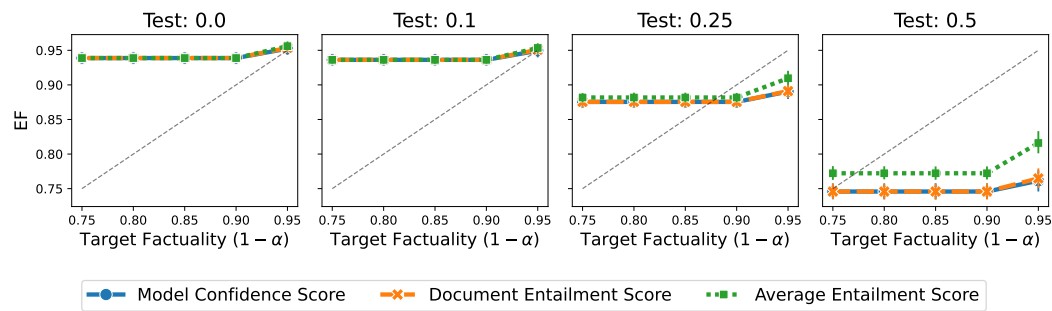

Figure 35: Empirical factuality under varying test hallucination proportions (0.0 to 0.5) with fixed clean calibration data. Three scoring methods compared on NQ-1K dataset with `Qwen3-4B`.

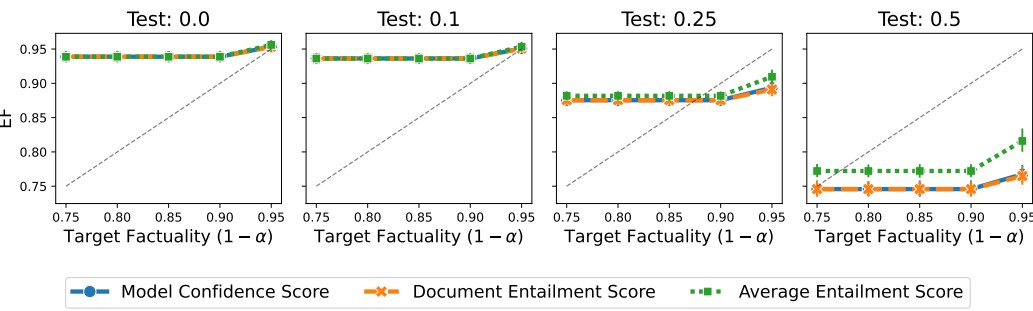

Figure 36: Empirical factuality under varying test hallucination proportions (0.0 to 0.5) with fixed clean calibration data. Three scoring methods compared on NQ-1K dataset with `Llama-3.2-3B-Instruct`.

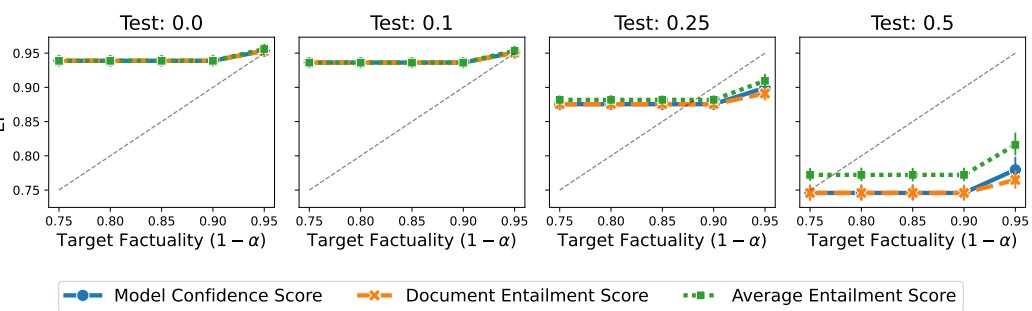

Figure 37: Empirical factuality under varying test hallucination proportions (0.0 to 0.5) with fixed clean calibration data. Three scoring methods compared on NQ-1K dataset with `SmolLM2-1.7B-Instruct`.

### A.2.7 ROBUSTNESS WITH ADVERSARIAL PREPARED CALIBRATION

We show the result for a setting with a test set with a distractor proportion of 0.25 and varying the level of distractors in the calibration set. Figure 38-45 illustrates that for all three datasets, FActScore, MATH-1K, and NQ-1K, when the fractions of distractors are underestimated, the empirical factuality (EF) is still below the target EF. The EF reaches the target EF only when we introduce a large enough fraction of distractors to the calibration set. However, we note that this incurs a high cost on the non-empty rate.

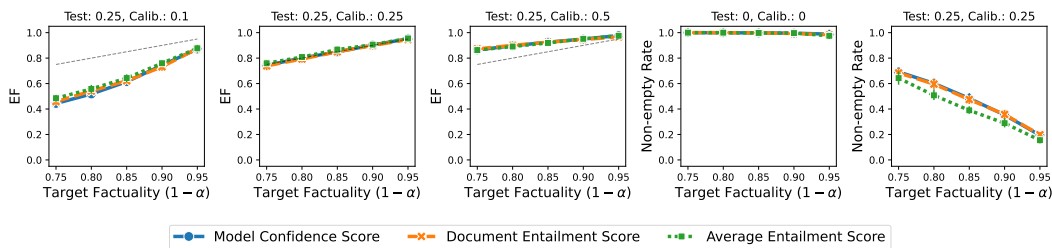

Figure 38: Empirical factuality and non-empty rates under different test-calibration distribution shifts. FActScore dataset with `Llama-3.2-3B-Instruct`. Shows trade-off between factuality guarantees and prediction coverage.

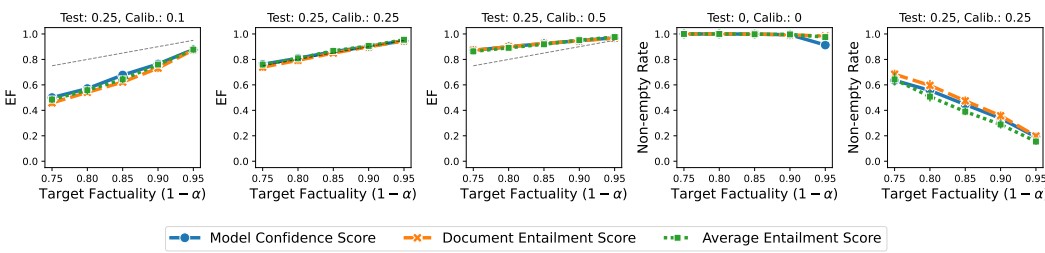

Figure 39: Empirical factuality and non-empty rates under different test-calibration distribution shifts. FActScore dataset with `SmolLM2-1.7B-Instruct`.

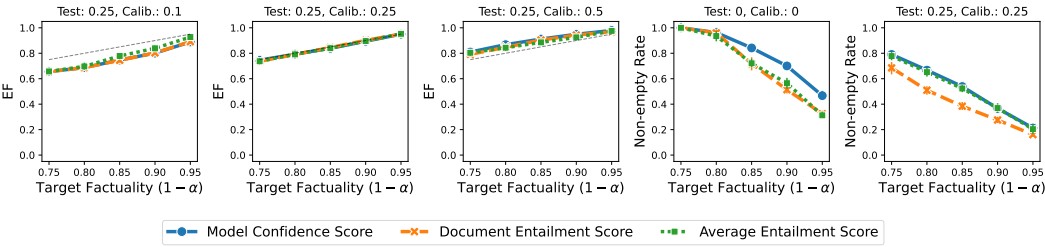

Figure 40: Empirical factuality and non-empty rates under different test-calibration distribution shifts. MATH-1K dataset with `Qwen3-4B`.

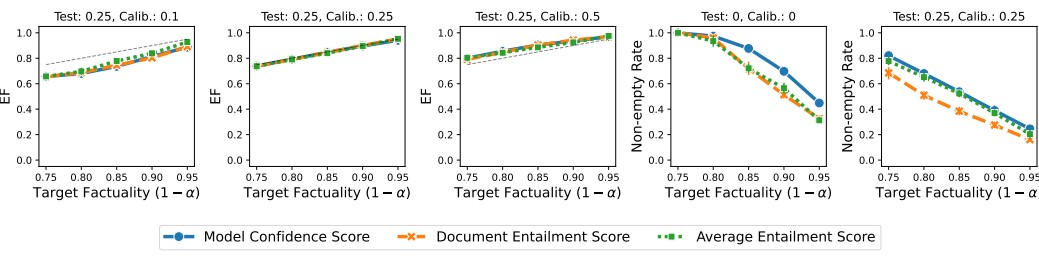

Figure 41: Empirical factuality and non-empty rates under different test-calibration distribution shifts. MATH-1K dataset with `Llama-3.2-3B-Instruct`.

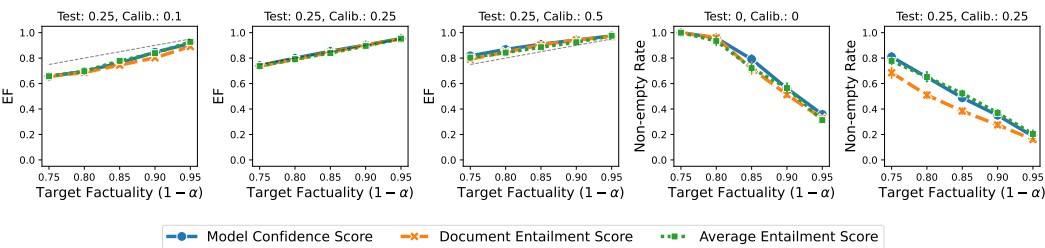

Figure 42: Empirical factuality and non-empty rates under different test-calibration distribution shifts. MATH-1K dataset with `SmolLM2-1.7B-Instruct`.

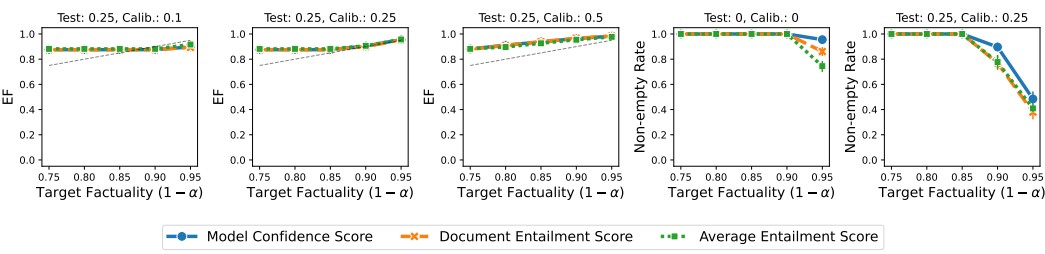

Figure 43: Empirical factuality and non-empty rates under different test-calibration distribution shifts. NQ-1K dataset with `Qwen3-4B`.

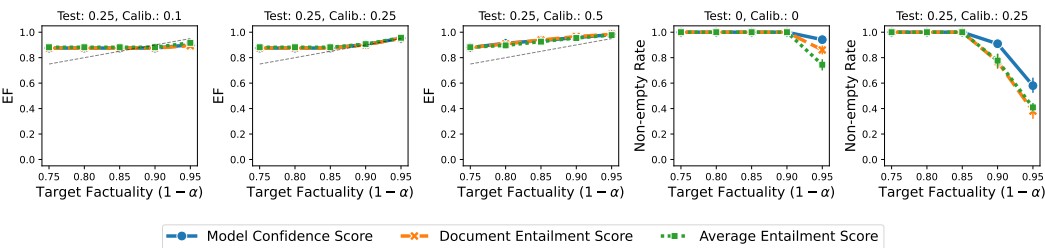

Figure 44: Empirical factuality and non-empty rates under different test-calibration distribution shifts. NQ-1K dataset with `Llama-3.2-3B-Instruct`.

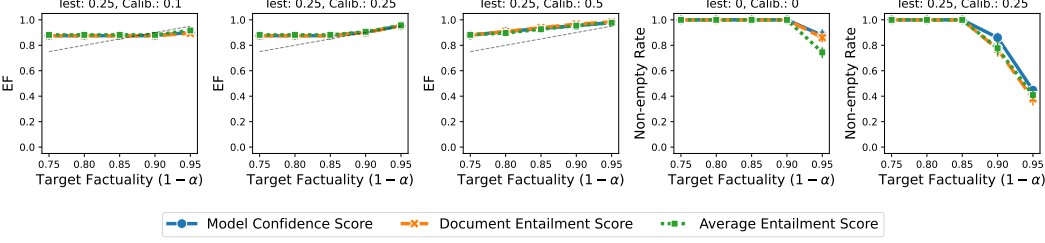

Figure 45: Empirical factuality and non-empty rates under different test-calibration distribution shifts. NQ-1K dataset with `SmolLM2-1.7B-Instruct`.

### A.2.8 END-TO-END EVALUATION

While our earlier analyses isolate the contributions of individual components—such as references for generation, scoring function design, and robustness under distribution shifts—a complete understanding requires evaluating the entire pipeline as a whole. Component-level results alone can be

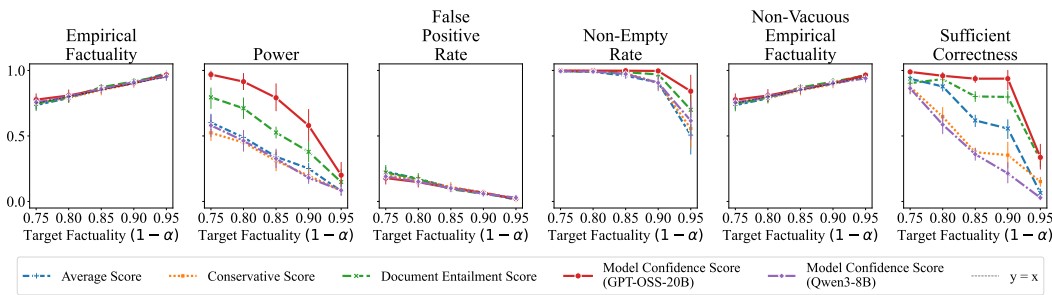

Figure 46: Performance of entailment- and confidence-based scorers using `gpt-oss-20b` and `Qwen3-8B` on FActScore.

misleading: a scorer may appear highly accurate in isolation but overly conservative when applied in practice, or references may improve raw generation quality but introduce new burdens on factuality filtering. Hence, now evaluate the full end-to-end pipeline. We use `gpt-oss-20b`, a mixture-of-experts model with 3.6B active parameters, as the response generator $G$ and consider two options for the scoring function $f$: (i) using `gpt-oss-20b` as the scorer, which reflects the practical scenario in which the same model is available for generation and scoring; (ii) using `Qwen3-8B` as the scorer, which allows us to study whether a dense model can serve as an alternative to a FLOPs efficient MoE model for scoring.

We report results on FActScore, MATH, and NQ, evaluating both factuality and correctness. In addition, we analyze computational cost by comparing floating-point operation counts (FLOPs) required for inference across the two models.

| Model | Total Tokens | Total FLOPs |
|---|---|---|
| `gpt-oss-20b` (3.6B active) (Agarwal et al., 2025) | 2000 | $1.44 \times 10^{13}$ |
| `Qwen3-8B` (8.19B active) (Yang et al., 2025) | 2000 | $3.28 \times 10^{13}$ |
| `DeepSeek-R1` (37B active) (DeepSeek-AI, 2025) | 2000 | $1.5 \times 10^{14}$ |
| `DeBERTa` (184M active) (He et al., 2020) | 2000 | $4.9 \times 10^{11}$ |
| `RoBERTa` (356M active) (Liu et al., 2019) | 2000 | $1.6 \times 10^{12}$ |

Table 1: Estimated FLOPs for generating 1000 tokens with a 1000-token prompt (assuming KV caching).

**Results and Discussion.** Figure 46 shows that confidence-based scoring with `gpt-oss-20b` achieves higher power and sufficient correctness than with `Qwen3-8B`, especially at moderate target factuality levels. Despite activating fewer parameters, `gpt-oss-20b` consistently outperforms `Qwen3-8B`, suggesting advantages from the MoE architecture. Table 1 further highlights the efficiency gap: `gpt-oss-20b` requires less than half the FLOPs of `Qwen3-8B` for comparable workloads, yet delivers stronger factuality filtering. We also note that the document entailment score, which is based on `DeBERTa`, is almost more than two orders of magnitude efficient than `gpt-oss-20b`, performs similarly to the model confidence score on the non-empty rate and sufficient correctness.

Together, these results demonstrate that parameter-efficient MoE models can be both more effective and more economical than dense counterparts for end-to-end factuality evaluation.

A.3 DISTRACTORS

A major assumption of the conformal factuality framework is that the test datapoints are exchangeable with the calibration dataset. However, in a real-world scenario where we deploy the model, it is possible that this assumption is mildly violated.

To test how robust these scoring functions are under this scenario, for each query $x_i$, the corresponding reference text $R(x)$, and the set of claims $\{c_i\}$, we ask the LLM to modify each $c_i$, given $x$ and

$R(x)$ such that it would become something that the model would hallucinate. We defer the prompts we used to generate these hallucination claims in Appendix A.5.6.

Our goal in creating these hallucination claims is that we want these claims to confuse the model so that the model would think that these claims are actually generated by them. To achieve this goal, after we generate a hallucinated claim, we ask the LLM to check if it thinks that the claim might be generated or hallucinated by itself (prompt in Appendix A.5.7), given the same $x, R(x)$. If the hallucination claim can cause the model to think that it is the one who generates it, given $x$ and $R(x)$, then we keep this hallucination claim. Otherwise, we repeat this process and generate a new hallucination claim.

## A.4    HUMAN EVALUATION

We conducted a human evaluation by randomly sampling 200 claims from the FActScore dataset. We had two students (referred to as A and B) label the factuality of these claims independently. We also used `gpt-5-nano` to label the same set of claims.

We then compared the agreement rates: `gpt-5-nano` vs. A, `gpt-5-nano` vs. B, and A vs. B. Table 2 shows the agreement rates. We found that the agreement rate between `gpt-5-nano` and a human labeler was around 75%, which was very similar to the inter-human agreement rate (A vs. B). This result suggests that using a capable LLM as a judge for factuality is a reasonable proxy, performing comparably to a human judge.

| Pair | Labelers | Agreement Rate |
|---|---|---|
| Model–Human | `gpt-5-nano` vs. Student A | 76.5% |
| Model–Human | `gpt-5-nano` vs. Student B | 77.0% |
| Human–Human | Student A vs. Student B | 73.0% |

Table 2: Agreement rates on factuality labels between `gpt-5-nano` and two human annotators, as well as between the two human annotators themselves.

## A.5    PROMPTS

### A.5.1    GENERATOR (WITH REFERENCE)

```
You are a helpful assistant that answers queries strictly based on the
    provided reference text.

Instructions:
1. You will be given:
   - A reference text
   - A query
2. Use only the information from the reference text to answer the query.
3. Do not include any information not supported by the reference text.

Output Requirements:
- Output ONLY a single VALID JSON5 object with EXACTLY this schema:
  {
     "response": "...answer strictly based on the reference text..."
  }

JSON5 Rules:
- Use DOUBLE QUOTES (") for all keys and all string values.
- Escape double quotes inside string values as \".
- Escape backslashes as \\.
- No trailing commas in objects or arrays.
- Use the exact top-level container specified and close it properly.
- Do not include comments, code fences, or any text outside the JSON5
    output.
- Follow the schema exactly; do not add or omit keys.
```

```
Do NOT include:
- Any text, explanations, comments, or formatting outside of the JSON5.
- Any code block delimiters (e.g., ```json).

Input:
Reference Text: {reference}
Query: {query}

(Reiteration of the instruction)
Answer the query strictly using only the reference text, and return a
    single JSON5 object with the key "response" only.

Output:
```

### A.5.2 GENERATOR (WITHOUT REFERENCE)

```
You are a helpful assistant that answers queries.

Instructions:
1. You will be given:
    - A query

Output Requirements:
- Output ONLY a single VALID JSON5 object with EXACTLY this schema:
    {
      "response": "...answer..."
    }

JSON5 Rules:
- Use DOUBLE QUOTES (") for all keys and all string values.
- Escape double quotes inside string values as \".
- Escape backslashes as \\.
- No trailing commas in objects or arrays.
- Use the exact top-level container specified and close it properly.
- Do not include comments, code fences, or any text outside the JSON5
    output.
- Follow the schema exactly; do not add or omit keys.

Do NOT include:
- Any text, explanations, comments, or formatting outside of the JSON5.
- Any code block delimiters (e.g., ```json).

Input:
Query: {query}

(Reiteration of the instruction)
Answer the query and return a single JSON5 object with the key "response"
    only.

Output:
```

### A.5.3 PARSER

```
You are an AI assistant tasked with breaking down input text into small,
    self-contained claims for easy human verification.

Instructions:
1. Parse the provided text into concise, independent, and non-overlapping
    subclaims.
2. Ensure each subclaim is:
  - As small and specific as possible.
  - Independent and self-contained.
```

```
1674    - Do not use pronouns like he, she, his, her, it, its, etc.
1675    - Explicitly mention subjects.
1676    - Factually complete without relying on context from other subclaims.
1677  3. If the provided text is not a full sentence, use the provided text
1678      verbatim as the subclaim
1679
1680  Output Requirements:
1680  1. The result must be a VALID and COMPLETE JSON list of dictionaries.
1681  2. Each dictionary must have the following structure:
1682     {
1683         "subclaim": "Subclaim text"
1683     }
1684
1685  JSON Rules:
1686  - Ensure the JSON is STRICTLY VALID:
1687    - Use DOUBLE QUOTES ("") for all keys and string values.
1688    - DO NOT include trailing commas after the LAST item in arrays or
1689      objects.
1690    - Ensure ALL dictionaries are enclosed in curly braces {}.
1691    - Ensure the JSON list is ENCLOSED in square brackets [].
1691    - CLOSE the JSON list properly with a closing square bracket ].
1692  - DO NOT include any code block delimiters.
1693  - DO NOT include any additional text, explanations, or comments in the
1694      outputreturn ONLY the JSON list.
1695  Critical Requirement:
1695  - Validate and ensure the JSON is complete, properly CLOSED, and
1696      parseable. The final JSON MUST pass validation without errors.
1697
1698  (Reiteration of the instruction)
1698  1. Parse the provided text into concise, independent, and non-overlapping
1699       subclaims.
1700  2. Ensure each subclaim is:
1701    - As small and specific as possible.
1702    - Independent and self-contained.
1703    - Do not use pronouns like he, she, his, her, it, its, etc.
1704    - Explicitly mention subjects.
1704    - Factually complete without relying on context from other subclaims.
1705  3. If the provided text is not a full sentence, use the provided text
1706      verbatim as the subclaim
1707
1708  Input:
1709  {input}
1710
1710  Output:
```

### A.5.4  LABELER (WITHOUT GROUND-TRUTH ANSWER)

```
   You are an AI assistant tasked with assigning a label to a claim based
      on its factuality.

  Instructions:
  1. You are given:
     - A reference text for the query.
     - A query.
     - A claim made in response to the query.
  2. Rate the factuality of the claim as a boolean:
     - true -> The claim is well-supported by the reference text.
     - false -> The claim contradicts the reference text, or is not well-
      supported by it.
     - If it is ambiguous, answer False.
  3. Provide:
     - The part(s) of the reference text that directly support your
      decision.
     - A reasoning statement describing your rationale.
```

```
4. You must assign either true or false. Never return null or None.

Output Requirements:
- Output ONLY a single VALID JSON5 object with EXACTLY these keys:
    {
      "highlighted_text": "Part(s) of the reference text that support the
      decision.",
      "reasoning": "A reasoning statement describing your rationale.",
      "answer": true
    }
- "answer" must be a boolean (true/false).

JSON5 Rules:
- Use DOUBLE QUOTES (") for all keys and all string values.
- Escape double quotes inside string values as \".
- Escape backslashes as \\.
- No trailing commas in objects or arrays.
- Use the exact top-level container specified and close it properly.
- Do not include comments, code fences, or any text outside the JSON5
    output.
- Follow the schema exactly; do not add or omit keys.

Do NOT include:
- Any text, explanations, comments, or formatting outside of the JSON5.
- Any code block delimiters (e.g., ```json).

Examples:
Example Input:
Reference Text: Michael Scott is a fictional character in the NBC sitcom
    The Office, portrayed by Steve Carell. Michael is the regional
    manager of the Scranton, Pennsylvania branch of Dunder Mifflin, a
    paper company, for the majority of the series. Like his counterpart
    in the original British version of the show, David Brent, he is
    characterized as a largely incompetent, unproductive, unprofessional
    boss, though he is depicted as kinder and occasionally shown to be
    effective at his job in key moments.
Query: Tell me a paragraph bio of Michael Scott.
Claim: The fictional character Michael Scott is the regional manager of a
    paper company.

Example Output:
{
  "highlighted_text": "Michael Scott is a fictional character in the NBC
    sitcom The Office, portrayed by Steve Carell. Michael is the regional
    manager of the Scranton, Pennsylvania branch of Dunder Mifflin, a
    paper company, for the majority of the series.",
  "reasoning": "Reference explicitly states Michael is regional manager
    at a paper company.",
  "answer": true
}

Example Input:
Reference Text: Michael Scott is a fictional character in the NBC sitcom
    The Office, portrayed by Steve Carell. Michael is the regional
    manager of the Scranton, Pennsylvania branch of Dunder Mifflin, a
    paper company, for the majority of the series. Like his counterpart
    in the original British version of the show, David Brent, he is
    characterized as a largely incompetent, unproductive, unprofessional
    boss, though he is depicted as kinder and occasionally shown to be
    effective at his job in key moments.
Query: Tell me a paragraph bio of Michael Scott.
Claim: The portrayal of Michael Scott in the NBC sitcom The Office is
    similar to that of David Brent, in the British version.

Example Output:
```

```
{
  "highlighted_text": "Like his counterpart in the original British
    version of the show, David Brent, he is characterized as a largely
    incompetent, unproductive, unprofessional boss, though he is depicted
     as kinder and occasionally shown to be effective at his job in key
    moments.",
  "reasoning": "Reference compares Michael Scotts characterization to
    David Brents, indicating similarity.",
  "answer": true
}

Example Input:
Reference Text: Michael Scott is a fictional character in the NBC sitcom
    The Office, portrayed by Steve Carell. Michael is the regional
    manager of the Scranton, Pennsylvania branch of Dunder Mifflin, a
    paper company, for the majority of the series. Like his counterpart
    in the original British version of the show, David Brent, he is
    characterized as a largely incompetent, unproductive, unprofessional
    boss, though he is depicted as kinder and occasionally shown to be
    effective at his job in key moments.
Query: Tell me a paragraph bio of Michael Scott.
Claim: Michael Scott is the founder of The Michael Scott Paper Company.

Example Output:
{
  "highlighted_text": "Michael is the regional manager of the Scranton,
    Pennsylvania branch of Dunder Mifflin, a paper company, for the
    majority of the series.",
  "reasoning": "Reference mentions only Dunder Mifflin; founding another
    company is not supported.",
  "answer": false
}

Example Input:
Reference Text: Michael Scott is a fictional character in the NBC sitcom
    The Office, portrayed by Steve Carell. Michael is the regional
    manager of the Scranton, Pennsylvania branch of Dunder Mifflin, a
    paper company, for the majority of the series. Like his counterpart
    in the original British version of the show, David Brent, he is
    characterized as a largely incompetent, unproductive, unprofessional
    boss, though he is depicted as kinder and occasionally shown to be
    effective at his job in key moments.
Query: Tell me a paragraph bio of Michael Scott.
Claim: Michael Scott is the CEO of Dunder Mifflin.

Example Output:
{
  "highlighted_text": "Michael is the regional manager of the Scranton,
    Pennsylvania branch of Dunder Mifflin, a paper company, for the
    majority of the series.",
  "reasoning": "Reference states regional manager, not CEO.",
  "answer": false
}

Input:
Reference Text: {reference}
Query: {query}
Claim: {claim}

(Reiteration of the instruction)
Return a single JSON5 object with "highlighted_text", "reasoning", and "
    answer" (true if supported; false if contradicted or unsupported).
    Assign true or falsenever null/None.

Output:
```

### A.5.5 LABELER (WITH GROUND-TRUTH ANSWER)

```
You are an AI assistant tasked with assigning a label to a claim based on
    its factuality.

Instructions:
1. You are given:
   - A reference text for the query.
   - A query.
   - A provided solution (final answer)
   - A claim made in response to the query.
2. Rate the factuality of the claim as a boolean:
   - true  The claim is well-supported by the reference text or match the
    given provided solution (final answer).
   - false  The claim contradicts the reference text or is not well-
    supported by it and contradicts the provided solution (final answer).
3. Provide:
   - The part(s) of the reference text or the provided solution that
    directly support your decision.
   - A reasoning statement describing your rationale.
4. You must assign either true or false. Never return null or None.

Output Requirements:
- Output ONLY a single VALID JSON5 object with EXACTLY this schema:
  {
    "highlighted_text": "Part(s) of the reference text or the provided
    solution that directly support the decision.",
    "reasoning": "A reasoning statement describing your rationale.",
    "answer": true
  }
- "answer" must be a boolean (true/false).

JSON5 Rules:
- Use DOUBLE QUOTES (") for all keys and all string values.
- Escape double quotes inside string values as \".
- Escape backslashes as \\.
- No trailing commas in objects or arrays.
- Use the exact top-level container specified and close it properly.
- Do not include comments, code fences, or any text outside the JSON5
    output.
- Follow the schema exactly; do not add or omit keys.

Do NOT include:
- Any text, explanations, comments, or formatting outside of the JSON5.
- Any code block delimiters (e.g., ```json).

Examples:
Example Input:
Reference Text: "Paris is the capital of France."
Answer: "Paris"
Query: "What is the capital of France?"
Claim: "Paris is the capital of France."

Example Output:
{
  "reasoning": "Claim matches reference and provided solution.",
  "answer": true
}

Example Input:
Reference Text: "Michael is the regional manager of Dunder Mifflin."
Answer: "Regional manager"
```

```
Query: "What is Michaels role at Dunder Mifflin?"
Claim: "Michael is the CEO of Dunder Mifflin."

Example Output:
{
  "reasoning": "Claim contradicts the reference and solution.",
  "answer": false
}

Example Input:
Reference Text: "Compute 12  3 + 4."
Answer: "40"
Query: "Evaluate 12  3 + 4."
Claim: "The value is 40."

Example Output:
{
  "reasoning": "Computation equals 40; claim matches the solution.",
  "answer": true
}

Example Input:
Reference Text: "Compute 12  3 + 4."
Answer: "40"
Query: "Evaluate 12  3 + 4."
Claim: "The value is 42."

Example Output:
{
  "reasoning": "The solution gives 40, not 42.",
  "answer": false
}

Example Input:
Reference Text: "The sum of the interior angles of a triangle is 180
    degrees."
Answer: "180 degrees"
Query: "What is the sum of the interior angles of a triangle?"
Claim: "The sum is 180 degrees."

Example Output:
{
  "reasoning": "Claim agrees with the stated geometric fact and the
    solution.",
  "answer": true
}

Input:
Reference Text: {reference}
Answer: {answer}
Query: {query}
Claim: {claim}

(Reiteration of the instruction)
Return a single JSON5 object with "highlighted_text", "reasoning", and "
    answer" (true if supported; false if contradicted or unsupported).
    Assign true or falsenever null/None.

Output:
```

### A.5.6   ATTACKER

```
You are a helpful assistant that modifies a correct claim into a false
    but plausible claim (hallucination) given a reference text and a
    query.

Instructions:
1. Read the reference text, the query, the original claim, and the two
    lists of prior modified claims (that do not work / that do work).
2. Generate exactly one false claim that:
    - Directly answers the query.
    - Contradicts or is not supported by the reference text.
    - Appears plausible given the reference and query (e.g., near-miss
    number/date, closely related entity, role swap).
    - Does not match or paraphrase any claim in either provided list.
3. Provide a reasoning statement describing why the false claim might
    look plausible.

Output Requirements:
- Output ONLY a single VALID JSON5 object with EXACTLY this schema:
    {
      "reasoning": "A reasoning statement describing why the false claim
      might look plausible.",
      "subclaim": "One-sentence false but plausible answer to the query."
    }
- The "subclaim" must be a standalone sentence that answers the query and
    is false with respect to the reference.

JSON5 Rules:
- Use DOUBLE QUOTES (") for all keys and all string values.
- Escape double quotes inside string values as \".
- Escape backslashes as \\.
- No trailing commas in objects or arrays.
- Use the exact top-level container specified and close it properly.
- Do not include comments, code fences, or any text outside the JSON5
    output.
- Follow the schema exactly; do not add or omit keys.

Do NOT include:
- Any text, explanations, comments, or formatting outside of the JSON5.
- Any code block delimiters (e.g., ```json).

Examples:
Example Input:
Modified claims that do not work: []
Modified claims that do work: []
Reference Text: "Gustave Eiffel oversaw the construction of the Eiffel
    Tower, which opened in 1889. The towers design is credited to Gustave
     Eiffel."
Query: "Who designed the Eiffel Tower?"
Claim: "Gustave Eiffel designed the Eiffel Tower."

Example Output:
{
  "reasoning": "Reference credits Eiffel; another associated architect
    name may look credible.",
  "subclaim": "Stephen Sauvestre designed the Eiffel Tower."
}

Example Input:
Modified claims that do not work: []
Modified claims that do work: []
Reference Text: "The capital of Australia is Canberra."
Query: "What is the capital of Australia?"
Claim: "Canberra."

Example Output:
```

```
{
  "reasoning": "Larger city misconception is common.",
  "subclaim": "Sydney is the capital of Australia."
}

Input:
Modified claims that do not work: {claims_not_work}
Modified claims that do work: {claims_work}
Reference Text: {reference}
Query: {query}
Claim: {claim}

(Reiteration of the instruction)
Generate exactly one false but plausible claim answering the query,
    unsupported by or contradicting the reference, not overlapping with
    provided lists; then return a JSON5 object with "reasoning" and "
    subclaim".

Output:
```

### A.5.7 CONFUSEE

```
You are a helpful assistant that evaluates whether a claim sounds like
    something you (an LLM) would plausibly sayand potentially
    hallucinatebased on the provided reference text and query.

Instructions:
1. Read the reference text, the query, and the claim.
2. Decide whether the claim is a plausible LLM-style statement that could
    arise from the reference and query even if it is not supported or is
    false.
3. Return true if the claim is plausibly hallucinated; otherwise return
    false.
4. Provide a reasoning statement describing your rationale.

Output Requirements:
- Output ONLY a single VALID JSON5 object with EXACTLY this schema:
  {
    "reasoning": "A reasoning statement describing your rationale.",
    "answer": (true|false)
  }
- "answer" must be a boolean (true/false).

JSON5 Rules:
- Use DOUBLE QUOTES (") for all keys and all string values.
- Escape double quotes inside string values as \".
- Escape backslashes as \\.
- No trailing commas in objects or arrays.
- Use the exact top-level container specified and close it properly.
- Do not include comments, code fences, or any text outside the JSON5
    output.
- Follow the schema exactly; do not add or omit keys.

Do NOT include:
- Any text, explanations, comments, or formatting outside of the JSON5.
- Any code block delimiters (e.g., ```json).

Examples:
Example Input:
Reference Text: "The capital of Australia is Canberra."
Query: "What is the capital of Australia?"
Claim: "Sydney is the capital of Australia."

Example Output:
```

```
{
  "reasoning": "Common misconception favors the largest city over the
    capital.",
  "answer": true
}

Example Input:
Reference Text: "Paris is the capital of France."
Query: "What is the capital of France?"
Claim: "Paris is the capital of France."

Example Output:
{
  "reasoning": "Claim is directly supported by the reference; not a
    hallucination.",
  "answer": false
}

Input:
Reference Text: {reference}
Query: {query}
Claim: {claim}

(Reiteration of the instruction)
Return a JSON5 object with "reasoning" and "answer" (true if plausibly
    hallucinated; else false).

Output:
```

### A.5.8   MODEL CONFIDENCE SCORE

```
You are an AI assistant tasked with assigning a confidence score to a
    claim based on its factuality.

Instructions:
1. You are given:
   - A reference text for the query.
   - A query.
   - A claim made in response to the query.
2. Rate the factuality of the claim with a numeric score in [0.0, 1.0]:
   - 0.0 -> The claim is false, contradicts the reference, or is not well
   -supported.
   - 1.0 -> The claim is factual and well-supported by the reference, or
   can be logically deduced from the reference and query.
3. Provide:
   - A score.
   - The parts of the reference text that directly support your scoring
   decision.
   - A reasoning statement describing your rationale.
4. You must assign a numeric score. Never return null, None, or a non-
   numeric value.

Important: A claim should be considered true (score 1.0) if it is either:
- Directly stated in the reference text, OR
- Can be logically deduced or calculated from the information provided in
    the reference text and query.
For mathematical claims, perform the necessary calculations based on the
    given data.
Output Requirements:
- Output ONLY a single VALID JSON5 object with EXACTLY these keys:
   {
     "highlighted_text": "Part(s) of the reference text that support the
     decision.",
     "reasoning": "A reasoning statement describing your rationale.",
```

```
        "score": 0.0-1.0
      }

  JSON5 Rules:
  - Use DOUBLE QUOTES (") for all keys and all string values.
  - Escape double quotes inside string values as \".
  - Escape backslashes as \\.
  - No trailing commas in objects or arrays.
  - Follow the schema exactly.

  Do NOT include:
  - Any text, explanations, comments, or formatting outside of the JSON5.

  Input:
  Reference Text: {reference}
  Query: {query}
  Claim: {claim}

  Reiteration of Instructions:
  Return a single JSON5 object with "highlighted_text", "reasoning", "score
      ". Assign a numeric scorenever null/None.

  Output:
```

### A.5.9 MERGER (FACTSCORE)

```
You will get an instruction and a set of facts that are true. Construct
    an answer using ONLY the facts provided, and try to use all facts as
    long as its possible. If the input facts are empty, output the empty
    string. Do not repeat the instruction.
Input:
The facts: {claims}
The instruction: {query}
Remember, If the input facts are empty, output the empty string. Do not
    repeat the instruction.
Output:
```

### A.5.10 MERGER (NATURAL QUESTIONS)

```
You will get a natural question and parts of an answer, which you are to
    merge into coherent prose. Make sure to include all the parts in the
    answer. There may be parts that are seemingly unrelated to the others
    , but DO NOT add additional information or reasoning to merge them.
    If the input parts are empty, output the empty string. Do not repeat
    the question.
Input:
The parts: {claims}
The question: {query}
Remember, DO NOT add any additional information or commentary, just
    combine the parts. If the input parts are empty, output the empty
    string. Do not repeat the question.
Output:
```

### A.5.11 MERGER (MATH)

```
You will get a math problem and a set of steps that are true. Construct
    an answer using ONLY the steps provided. Make sure to include all the
     steps in the answer, and do not add any additional steps or
    reasoning. These steps may not fully solve the problem, but merging
    them could assist someone in solving the problem. If the input steps
    are empty, output the empty string. Do not repeat the math problem.
```

```
Input:
The steps: {claims}
The math problem: {query}
Remember, do not do any additional reasoning, just combine the given
    steps. If the input steps are empty, output the empty string. Do not
    repeat the math problem.
Output:
```

### A.5.12  CORRECTNESS

```
I need your help in evaluating an answer provided by an LLM against
    ground truth answers. Your task is to determine if the LLMs response
    matches the ground truth answers. Please analyze the provided data
    and make a decision.

Instructions:
1. Carefully compare the "Predicted Answer" with the "Ground Truth
    Answers".
2. Consider the substance of the answers  look for equivalent information
     or correct answers. Do not focus on exact wording unless the exact
    wording is crucial to the meaning.
3. Your final decision should be based on whether the meaning and the
    vital facts of the "Ground Truth Answers" are present in the "
    Predicted Answer."
4. Categorize the answer as one of the following:
- "perfect": The answer is completely correct and matches the ground
    truth.
- "acceptable": The answer is partially correct or contains the main idea
     of the ground truth.
- "incorrect": The answer is wrong or contradicts the ground truth.
- "missing": The answer is "I dont know", "invalid question", or similar
    responses indicating lack of knowledge.

Output Requirements:
- Output ONLY a single VALID JSON5 object with EXACTLY this schema:
   {
     "reasoning": "A reasoning statement describing your rationale.",
     "answer": "One of perfect, acceptable, incorrect, or missing"
   }

JSON5 Rules:
- Use DOUBLE QUOTES (") for all keys and all string values.
- Escape double quotes inside string values as \".
- Escape backslashes as \\.
- No trailing commas in objects or arrays.
- Use the exact top-level container specified and close it properly.
- Do not include comments, code fences, or any text outside the JSON5
    output.
- Follow the schema exactly; do not add or omit keys.

Do NOT include:
- Any text, explanations, comments, or formatting outside of the JSON5.
- Any code block delimiters (e.g., ```json).

Input:
Query: {query}
Predicted Answer: {merged_string}
Ground Truth Answer: {answer}

Reiteration of Instructions (before output):
1. Carefully compare the "Predicted Answer" with the "Ground Truth
    Answers".
```

```
2. Consider the substance of the answers  look for equivalent information
    or correct answers. Do not focus on exact wording unless the exact
    wording is crucial to the meaning.
3. Your final decision should be based on whether the meaning and the
    vital facts of the "Ground Truth Answers" are present in the "
    Predicted Answer."
4. Categorize the answer as one of the following:
- "perfect": The answer is completely correct and matches the ground
    truth.
- "acceptable": The answer is partially correct or contains the main idea
    of the ground truth.
- "incorrect": The answer is wrong or contradicts the ground truth.
- "missing": The answer is "I dont know", "invalid question", or similar
    responses indicating lack of knowledge.

Output:
```

### A.5.13 SUFFICIENT CORRECTNESS

```
You are an expert LLM evaluator that excels at evaluating a RESPONSE with
    respect to a QUERY given a REFERENCE.

Consider the following criteria:
Sufficient Correctness:
1 IF the RESPONSE contains a sufficient amount of CORRECT information (
    verified against the REFERENCE) to infer the answer to the QUERY.
0 IF the RESPONSE does not contain a sufficient amount of CORRECT
    information to infer the answer to the QUERY.
Important:
Judge only the correctness of the RESPONSE content according to the
    REFERENCE.
Do not infer correctness from external knowledge.

Output ONLY a single VALID JSON5 object with EXACTLY these keys:
{
  "explanation": An explanation describing your rationale, with with you
    will make your decision on sufficient_correctness,
  "sufficient_correctness": 1 IF the RESPONSE contains a sufficient
    amount of correct information (verified against the REFERENCE) to
    infer the answer to the QUERY, 0 IF the RESPONSE does not contain a
    sufficient amount of correct information to infer the answer to the
    QUERY.
}
- "sufficient_correctness" must be an integer (0/1).

JSON5 Rules:
- Use DOUBLE QUOTES (") for all keys and all string values.
- Escape double quotes inside string values as \".
- Escape backslashes as \\.
- No trailing commas in objects or arrays.
- Use the exact top-level container specified and close it properly.
- Do not include comments, code fences, or any text outside the JSON5
    output.
- Follow the schema exactly; do not add or omit keys.

Do NOT include:
- Any text, explanations, comments, or formatting outside of the JSON5.
- Any code block delimiters (e.g., ```json).

Input:
### QUERY
{query}
### REFERENCE
{reference}
```

```
### RESPONSE
{response}
Output:
```

### A.5.14   MATH REFERENCE

```
You are a helpful assistant that extracts the prerequisite mathematics
    knowledge needed to answer a given questionwithout solving it.

Instructions:
1. Read the question.
2. Identify only the minimal prerequisite items across concepts,
    definitions, theorems/properties, formulas, techniques, notation,
    assumptions/conditions, and common pitfalls required to answer the
    question.
3. When you state a theorem, a definition, or a formula, write out its
    full, standard statement (not just the name). For theorems, include
    hypotheses and conclusions; for definitions, give the precise meaning
    ; for formulas, write the exact equation(s) in standard notation.
4. Do NOT provide the answer or partial solution steps.
5. Do NOT use examples that arise directly from the given question; keep
    statements general and problem-agnostic.

Output Requirements:
- Produce PLAIN TEXT ONLY.
- Write free-form prose (no headings, lists, or numbering).
- Mention required topics, definitions, fully written theorems/properties
    , formulas, techniques, notation conventions, assumptions/conditions,
    and common pitfalls in sentences.
- The output may be long; include complete statements where needed.

Do NOT include:
- Any answer, hints, or step-by-step solution.
- Any examples derived from the given question.

Reiteration of the instructions:
1. Read the question.
2. Identify only the minimal prerequisite items across concepts,
    definitions, theorems/properties, formulas, techniques, notation,
    assumptions/conditions, and common pitfalls required to answer the
    question.
3. When you state a theorem, a definition, or a formula, write out its
    full, standard statement (not just the name). For theorems, include
    hypotheses and conclusions; for definitions, give the precise meaning
    ; for formulas, write the exact equation(s) in standard notation.
4. Do NOT provide the answer or partial solution steps.
5. Do NOT use examples that arise directly from the given question; keep
    statements general and problem-agnostic.

Input:
{query}

Output:
```

## B   ETHICS STATEMENTS

This work adheres to the ICLR Code of Ethics. We do not anticipate any direct risks of harm to individuals or groups arising from this research.

## C    REPRODUCIBILITY STATEMENT

We are committed to ensuring the reproducibility and transparency of our work. To this end, we will fully open-source our code upon publication. The released repository will include the complete implementation scripts. All datasets and models used in this study are already publicly available.

## D    USAGE OF LARGE LANGUAGE MODELS

Large language models were employed during the preparation of this manuscript. Commercial LLMs were used solely for editing purposes, improving clarity, grammar, and adherence to academic style, without altering the authors' intended meaning or substantive contributions.

