# OpenReview forum: "Understanding Conformal Factuality for RAG-based LLMs: Novel Metrics and Systematic Insights"
_ICLR.cc/2026/Conference — Submitted to ICLR 2026_

### Official Review · Reviewer_fFdw · 2025-10-27

**Soundness:** 3
**Presentation:** 3
**Contribution:** 2
**Rating:** 4
**Confidence:** 3

**Summary:**

In this work, the authors investigate a conformal factuality framework with RAG: the performance and the limitations. Three  metrics are introduced for standard factuality limitation.  According to the experiments on three datasets (FActScore, MATH, and NQ), importance of designing scoring functions is concluded.

**Strengths:**

1. The paper unifies two complementary paradigms (RAG + CP) and offers a principled framework for combining grounding and statistical guarantees. Comparing to previous work, a comprehensive understanding of strenghts and limitations is explored.

2. Comprehensive empirical design: evaluates across three diverse datasets (FActScore, MATH, Natural Questions) and multiple model families (Qwen3, SmolLM2, Llama-3.x), enabling cross-domain and cross-architecture study.

3. Analysis on robustness of scoring function: distribution shift and distractor. And some other effect is done: access to reference, prompt strategy, modeling scaling.

**Weaknesses:**

1. Limited novelty in the algorithmic contribution: It shows systematic analysis rather than a new algorithmic integration of RAG and CP.

2. In the study, score function is needed. Entailmment-based scores and LLM-based scores are mentioned. Reference is usually needed. The limitation add the challenging of real-word (noisy retrieval or insufficient information). In addition, although some comparison is done for the two family  scorers, it is not comprehensive: e.g,, larger model (>8B), other architecture (MOE) etc. And it is unclear why Entailmment-based scores are better.

3. The metrics: Lot of metrics do not change a lot. It seems only "Power" metric have a higher variance for different settings. There is no big variance for the proposed metrics (Non-empty Rate, Non-vacuous Empirical Factuality, Correctness, Sufficient Correctness). The value of the proposed metrics are unclear.

**Questions:**

1. Practical deployment:
In real applications (e.g., QA systems), how would users set the target factuality level 1−α? Is there guidance on selecting α to balance factuality and informativeness?

2. Entailment vs confidence scorers:
Why do entailment-based scorers sometimes outperform larger LLM-based ones? Is it due to explicit textual grounding or better calibration properties?

---

> ### Author Response · Authors · 2025-11-16
>
> We thank the reviewer for their thoughtful comments and questions. We'd like to address each point below:
>
> **Takeaways**: Our findings, particularly from Sections 4 and 5, offer several implications for building reliable information-seeking agents. For instance, our work highlights that relying on the evaluation of empirical factuality and power on well-calibrated benchmark datasets is not enough to capture the usefulness of the system. For example, an empty answer is by default factually correct; a factually correct statement that is unrelated to the query would not be distinguished from a factually correct statement that is related to the query while computing statistical power. Our metrics address this, allowing for better practical evaluation of the system.
>
> Furthermore, our extensive evaluations of various scoring functions show that lightweight entailment-based models can match or even outperform much larger LLMs. See Table 1 in Appendix A.2.8 for the FLOPs required for these models to make an inference. Our results reveal that entailment-based models are a better choice for scoring functions in a practical setting.
>
> Lastly, as we note the exchangeability assumption needed for conformal prediction guarantees is often violated in practice, i.e., there is a distribution shift between calibration and test set, the statistical guarantee provided by conformal filtering no longer holds. This highlights the need for focusing on robustness from the ground up. We envision this will motivate researchers to focus on developing robust scoring functions and to also fundamentally re-think frameworks for factuality guarantees beyond conformal filtering.
>
> **On Why Entailment-based Scorers are Better**:  The entailment-based models are specifically fine-tuned for assessing whether a reference text supports a given claim [1][2]. On the other hand, the LLM-based scorers are general-purpose generative models; we are prompting them to perform a complex, zero-shot scoring task. This could be a reason for better performance by entailment-based scorers. This suggests that the computationally more efficient entailment models are more suitable for serving as a scoring function.
>
> [1] https://arxiv.org/abs/1907.11692
>
> [2] https://huggingface.co/MoritzLaurer/DeBERTa-v3-base-mnli-fever-docnli-ling-2c

---

> ### Author Response · Authors · 2025-11-16
>
> **Our Contribution**: We respectfully disagree with the premise that a systematic analysis is a lesser contribution.
>
> We would like to re-emphasie our contributions:
> 1. To our knowledge, our work is the **first** to perform a **comprehensive systematic evaluation of conformal factuality for RAG-based LLM systems**—across diverse tasks, model families, and scoring functions.
> 2. We propose **novel metrics for evaluation** designed to better capture the “usefulness” of conformal factuality systems. These metrics complement existing ones and provide evaluation beyond traditional empirical factuality, aiming to address practical needs in real-world deployment.
> 3. Systematic **evaluation framework focusing on robustness to distribution shifts**, a key aspect for practical considerations. These contribute actionable insights to both practitioners and researchers.
>
> These are foundational for practical and research purposes: Our novel metrics and evaluation framework is useful for anyone seeking to build a reliable LLM system. Our findings highlight the need for re-thinking reliability guarantees for RAG-based LLMs keeping robustness from the ground up – both towards better scoring functions in CP filtering and towards developing statistical methods beyond CP filtering.
>
> **On Retrieval**: The focus of the paper is on understanding **conformal filtering** for RAG-based LLMs. The key question is whether conformal factuality can ensure factuality in ideal settings while still preserving the usefulness of the final response. Assuming "oracle retriever” precisely allows us to focus on evaluating the effectiveness, usefulness, sensitivity, and robustness of the conformal factuality system (which is used for providing statistically meaningful guarantees on factuality of the final response).
>
> We would like to emphasize that we have conducted systematic experiments on different tasks (open-ended generation, mathematical reasoning, and question answering)  using models from various families (Qwen3, SmoLM2, Llama3), different model sizes and various scoring functions. These different combinations already lead to combinatorially many experiments.
>
> Studying strengths and limitations of the RAG is of independent interest and would make a complementary project. We would also like to highlight that trying to fit everything in a single work would be infeasible.
>
> **On Comprehensive Comparisons**: We respectfully disagree with the claim that our work is not comprehensive. Our systematic evaluation is conducted over multiple datasets and tasks (open-ended generation, mathematical reasoning, and question-answering) using multiple model families (Qwen3, SmolLM2, Llama3) and various scoring functions. Our experiments as a whole are already combinatorically complex.
>
> Due to page limitations, we already had to defer many results, e.g., the end-to-end evaluation to Appendix A.2.8, where we conducted experiments using gpt-oss-20b, a larger model with the mixture-of-experts architecture. We will add references to these experiments in the appendix in the main paper.
> We also ran additional experiments on the Qwen3-32B (and its 'Think' variant) and Qwen3-30B-A3B model in Section 4.3. We have updated the results in Figure 5. We observe that on the FActScore dataset, Qwen3-0.6B still performs the best across sufficient correctness. On the MATH dataset (Figure 27 in Appendix A.2.4), Qwen3-8B is competitive with its larger, 30B+ variants. These results align with our previous finding that simply scaling the model size does not always help.
>
> **On the Value of Proposed Metrics**: Our new metrics are essential for revealing these non-obvious dynamics. For example, in our updated Figure 5, it is the sufficient correctness metric that clearly shows Qwen3-0.6B performing better than its larger variants.
>
> Furthermore, the non-empty rate was critical in our robustness tests. As seen in Figure 9 (d) vs (e), when we add just 25% distractors to the calibration set, the non-empty rate collapses from a high-utility rate (near 1.0) to a low-utility one (plunging to ~0.25). This massive variance is the key signal. This metric is what allows us to quantify the pathological trade-off: in a noisy environment, the system is forced to become useless to stay "safe."
>
> We hope our response resolves your queries, and we hope that you consider increasing the score. We are happy to answer any further questions.

---

### Official Review · Reviewer_hF6H · 2025-11-01

**Soundness:** 1
**Presentation:** 3
**Contribution:** 2
**Rating:** 4
**Confidence:** 4

**Summary:**

This paper studies RAG with conformal factuality filtering, a framework designed to provide statistical guarantees on output factuality. The authors conduct a systematic analysis across diverse datasets (FactScore, MATH, NQ) and model families(Qwen3 0.6B, 4B, 8B; Llama-3.2 1B&3B; SmlLM2), investigating critical components such as scoring function design, the role of references, model capacity, and robustness to distractors. Key findings indicate that while providing references consistently improves generation and scoring, current conformal filtering methods are highly sensitive to distribution shifts and adversarial distractors, highlighting the need for more robust scoring functions to ensure both conformal factuality are followed.

**Strengths:**

1. I appreciate the systematic studies on conformal filtering in a practical RAG setup (Section 5), which matches much more with the realistic setting, ie distraction happens, and the limitation of calibration set's collection. This result reveals the practical challenges of conformal filtering's guarantee where imperfect retrieval doesn't exist.

2. They presents a good coverage of evaluation to carefully consider the weakness of Empirical Factuality, and supplement Non-empty Rate, Non-vacuous Empirical Factuality.

**Weaknesses:**

1.  There are couples of statement is unclear in this paper and several inconsistency within the paper's description. More specifically:
1.1. It's unclear on how the calibration set and claims are extracted (Line 97-98)
1.2. The results (except Figure 2), doesn't use Sufficient Correctness (SC) at all, but it were claimed as a novel evaluation metrics contribution
1.3. Line 190 claims gpt-oss-20b is used, but no any results related to this model.

2. The coverage of the model are very small size in general. Given nowadays the most powerful Proprietary LLMs are faithfully strong when reference is provided, how the conformal factuality filtering will work is a much worth study direction. Given the limited novelty of the paper, it's hard to credit the paper enough if the study coverages are without stronger and deeper understanding about the frontier models.

3. Could diver a bit deeper more on the practical side for the calibration cost: Conformal methods often require expensive calibration; the paper does not discuss computational or data efficiency trade-offs.

4. Missing human evaluation: While factual metrics are well-defined, it would helpful to further validate the alignment with human judgement.

**Questions:**

See weakness for questions.

Some additional suggestions:
1. Line 080: "strenghts" => "strength"

---

> ### Author Response · Authors · 2025-11-16
>
> We thank the reviewer for their constructive feedback and thoughtful questions, which have helped us strengthen our results. We have corrected the spelling issue on Line 80. Below, we address the reviewer's queries and offer some clarifications on our writing:
>
> **Clarifications**: We appreciate the opportunity to clarify these points:
>
> 1. The calibration set is a small, held-out set of 50 queries. For each query in this set, we generate a response, parse it into claims, and label them as factual or not. For a given scoring function, these claims are scored, and then we find the filtering threshold $\tau_{\alpha}$ to guarantee factuality at level $1-\alpha$ (see Section 2 for more detail).
> 2. We have updated the draft to include the sufficient correctness metric in Figure 5 and 6.
> 3. Figure 46 in the Appendix contains an end-to-end evaluation involving gpt-oss-20b. We will make it clear in the main paper that the results for gpt-oss-20b are in the Appendix.
>
> **On Model Coverage**: Retrieval augmented generative LLMs lack statistical guarantees on answers generated, which is needed in safely critical applications, e.g., summary of medical reports. This is exactly where conformal filtering has emerged as a dominant, promising approach to guarantee factuality.  Our study precisely focuses on a systematic evaluation of conformal filtering for RAG-based LLMs.  Figure 2 shows that even small and medium-sized LLMs, when equipped with good reference, can already generate responses that retain a good amount of information to answer the given query correctly. The main question is how efficient and robust the statistical framework of CP-based filtering is for guaranteeing factuality. That is, how useful and factually correct is the final output $y’$ in Figure 1. **We ran a new experiment with Gemini 2.5 Pro and GPT 5.1** to compare sufficient correctness in the initial response y (Figure 1) conditioning on the presence of a reference text. Figure 11 shows the results. We can see that Qwen3-4B is comparable to these frontier models when the reference text is given.
>
> Our observations thus would be relevant to providing guarantees for answers generated by the frontier models as well, e.g., a frontier model LLM can be used for generating the initial response $y$ (Figure 1). Additionally, we want to emphasize that using frontier LLMs in every part of the CP filtering system would be very costly. Table 1 in Appendix A.2.8 reports the FLOPs required for different models to generate text. As shown, large models such as DeepSeek-R1 remain roughly an order of magnitude more expensive than models of more modest size. Repeatedly using these frontier models for claim scoring would incur a huge cost.
>
> That said, **we also ran additional experiments on larger models: Qwen3-32B and Qwen3-30B-A3B**. (new results in Figure 5). We observe that on the FActScore dataset, Qwen3-0.6B still performs the best across sufficient correctness and power. On the MATH dataset (Figure 27 in Appendix A.2.4), Qwen3-8B is competitive with its larger, 30B+ variants. These results align with our previous finding that simply scaling the model size does not always help.
> Additionally, using models that are of modest size provides a viable path for developers, researchers, and organizations without access to large-scale computers to build reliable, factually-grounded systems.
>
> **Cost of Conformal Filtering**:
> From a computational cost perspective, the calibration process is highly efficient and not expensive computationally, as it is not an iterative training or fine-tuning process. It is a simple, one-shot "generate-score-sort" pipeline. On the other hand, conformal filtering is indeed expensive in terms of needing human labeled data. Our experiments show that even when we change the generator G (Figure 1). We may need to collect new calibration data in order to make sure the exchangeable assumption is not broken.
>
> **On Human Evaluation**: We conducted a human evaluation by randomly sampling 200 claims from the FActScore dataset. We had two students (A and B) label these claims independently. We also used gpt-5-nano to label the same set of claims.
>
> We then compared the agreement rates: gpt-5-nano vs. A, gpt-5-nano vs. B, and A vs. B. We found that the agreement rate between `gpt-5-nano` and a human labeler was around 80%, which was very similar to the inter-human agreement rate (A vs. B). This result suggests that using a capable LLM as a judge for factuality is a reasonable proxy, performing comparably to a human judge. See Appendix A.4 for more detailed results.
>
> |          Labelers        | Agreement Rate |
> |--------------------------|----------------|
> | GPT-5-nano vs. Student A | 76.5%          |
> | GPT-5-nano vs. Student B | 77.0%          |
> | Student A  vs. Student B | 73.0%          |
>
> We hope our response resolves your queries, and we hope that you consider increasing the score. We are happy to answer any further questions.

---

### Official Review · Reviewer_GfF5 · 2025-11-03

**Soundness:** 2
**Presentation:** 2
**Contribution:** 2
**Rating:** 2
**Confidence:** 3

**Summary:**

This paper investigates the integration of Retrieval-Augmented Generation (RAG) and the Conformal Factuality (CP) filtering framework to mitigate LLM hallucinations. The authors found that aggressive filtering to maintain factuality often leads to a high rate of empty or vacuous outputs. To evaluate this challenge, they introduce three new metrics—Non-empty Rate, Non-vacuous Empirical Factuality, and Sufficient Correctness—which explicitly capture the balance between correctness and informativeness (utility). Key results show that references significantly boost generation quality, lightweight entailment-based scoring functions are often more efficient and effective than large LLM confidence scorers, and the entire system is highly sensitive to distribution shifts and distractors in the data.

**Strengths:**

- The study is done in various datasets and LLMs families.
- The experiments show insightful findings like critical flaw in standard measures which fail to penalize empty or useless outputs, and vulnerability to distribution shifts and adversarial distractors.

**Weaknesses:**

- The paper contribution is limited. While the systematic analysis is appreciated, the core contribution is largely incremental, relying on the mechanistic combination of two established techniques.
- The study's findings are heavily focused on the trade-offs within the combined method (e.g., comparing different scorers or prompt designs for the combined system). However, a comprehensive analysis requires more robust comparison against strong standalone baselines under the exact same circumstances.

**Questions:**

Your robustness tests show a core conflict: forcing high accuracy (EF/NvEF) makes the output unhelpfully empty (low NR). For a practical RAG system that needs to be both trustworthy and useful, what specific combination of your new metrics (NvEF, NR, and SC) should a practitioner target? Is there a principled way to choose between a very safe but less informative answer (e.g., 90% correctness, 50% coverage) and a more complete answer that is slightly riskier (80% correctness, 80% coverage)?

---

> ### Author Response · Authors · 2025-11-16
>
> We thank the reviewer for their constructive feedback and thoughtful questions. They have helped us strengthen our results. We want to address the specific queries below:
>
> **Our Contribution**: Our work is the first to systematically study conformal filtering for RAG-based LLMs. In particular, our analysis of the sensitivity and robustness of CP filtering is novel. We also proposed novel metrics to evaluate the usefulness of the system. Our systematic evaluation is conducted over multiple types of tasks (open-ended generation, mathematical reasoning, and question-answering) using multiple model families and various scoring functions. Our experiments provide valuable insights into the strengths and limitations of conformal filtering that contribute actionable insights to both practitioners and researchers.
>
> So, we respectfully disagree with the characterization and dismissal of our contributions. There are different types of contributions in research. Limiting the scope of contributions to a very narrow set would be myopic.
>
> **Other Baseline**: We find this comment somewhat unclear and would appreciate further clarification. Specifically, could the reviewer elaborate on what is meant by a “robust comparison”? In addition, could you please clarify which “strong standalone baselines” you are referring to? We would appreciate it if you could point to the specific reference(s) you have in mind.
> To reiterate, our experiments already span multiple task types (open-ended generation, mathematical reasoning, and question answering), model families (Qwen3, SmolLM2, and Llama3), and scoring functions.
>
> **On Trade-offs between Different Metrics**: The trade-off between factuality and usefulness depends on the application.
>
> In particular, it hinges on the specific cost of a factually incorrect statement (a hallucination) versus the cost of not answering with useful information for a given task. For example, in a high-stakes healthcare setting, the tolerance of factually incorrect statements is very low. The goal here would be to guarantee a predetermined level of factual accuracy, e.g., 95% guaranteed factuality and try to maximize usefulness under this constraint. This would be a constrained optimization to search for the best combination of scoring function, LLMs, etc., keeping in mind computational considerations as well. In other applications, one could weigh different metrics differently and aim for multi-objective optimization.
>
> Our new metrics allow the practitioner to choose the metrics that are appropriate for the goal of the system they are building to make this cost-benefit analysis explicit and tune their system accordingly.
>
> Additionally, our evaluation setting provides a framework for analyzing and testing the robustness of the system before deployment.
>
> We hope our response resolves your queries, and we hope that you consider increasing the score. We are happy to answer any further questions.

---

### Official Review · Reviewer_tKcX · 2025-11-03

**Soundness:** 2
**Presentation:** 3
**Contribution:** 3
**Rating:** 2
**Confidence:** 4

**Summary:**

This paper systematically investigates the integration of Conformal Factuality (CF) filtering with Retrieval-Augmented Generation (RAG) to mitigate Large Language Model (LLM) hallucinations while providing statistical guarantees. The authors identify a critical limitation in existing CF metrics, where empty responses are considered perfectly factual. To address this, they introduce three new metrics: Non-empty Rate (NR), Non-vacuous Empirical Factuality (NvEF), and Sufficient Correctness (SC), which better balance factuality with informativeness.

The study conducts comprehensive experiments across multiple datasets (FActScore, MATH, Natural Questions) and model families (Qwen3, Llama-3.2, SmolLM2).

**Strengths:**

- The introduction of new metrics for measuring factuality makes sense.
- As an empirical study paper, it's good to consider multiple model families and benchmarks.

**Weaknesses:**

- While titled as "RAG-based," much of the core setup assumes an "oracle retriever" that always provides relevant references (Section 2), which is a bit unrealistic. The quality of the reference should be critical as well - beyond simple binary existence in Section 3.

- After reading the paper, I find it hard to connect the insights the authors obtained to real-world settings and their practical implications. For example, how we apply the findings to challenging agentic scenarios for designing effective information-seeking agents (based on RAG or Internet Search)?

- Why stronger models of larger sizes are not included in the study? For example, we have Qwen3-30B and its thinking variants, which are widely used in more practical agentic settings.

- Unfinished sentence in line 172.

- Margin issue of Figure 4 and Figure 8.

**Questions:**

See weakness.

---

> ### Author Response · Authors · 2025-11-16
>
> We thank the reviewer for the constructive feedback and questions that have helped strengthen our results. We have fixed the margin issue of Figure 4 and Figure 8, and the sentence in line 172. Below, we address the queries that the reviewer has:
>
> **On Oracle Retriever**: The focus of the paper is on understanding **conformal filtering** for RAG-based LLMs. The key question is whether conformal factuality can ensure factuality in ideal settings while still preserving the usefulness of the final response. Assuming "oracle retriever” precisely allows us to focus on evaluating the effectiveness, usefulness, sensitivity, and robustness of the conformal factuality system (which is used for providing statistically meaningful guarantees on factuality of the final response).
>
> We would like to emphasize that we have conducted systematic experiments on different tasks (open-ended generation, mathematical reasoning, and question answering)  using models from various families (Qwen3, SmoLM2, Llama3), different model sizes, and various scoring functions. These different combinations already lead to combinatorially many experiments.
>
> Studying strengths and limitations of the RAG is of independent interest and would make a complementary project. We would also like to highlight that trying to fit everything in a single work would be infeasible.
>
> **Takeaways**: Our findings, particularly from Sections 4 and 5, offer several implications for building reliable information-seeking agents. For instance, our work highlights that relying on evaluation of empirical factuality and power on well calibrated benchmark datasets is not enough to capture the usefulness of the system. For example, an empty answer is by default factually correct, a factually correct statement that is unrelated to the query would not be distinguished from a factually correct statement that is related to the query while computing statistical power. Our metrics address this allowing for better practical evaluation of the system.
>
> Furthermore, our extensive evaluations of various scoring functions show that lightweight entailment-based models can match or even outperform much larger LLMs. See Table 1 in Appendix A.2.8 for the FLOPs required for these models to make an inference. Our results reveal that entailment-based models are a better choice for scoring functions in a practical setting.
>
> Lastly, as we note the exchangeability assumption needed for conformal prediction guarantees is often violated in practice, i.e., there is a distribution shift between calibration and test set, the statistical guarantee provided by conformal filtering no longer holds. This highlights the need for focusing on robustness from the ground up. We envision this will motivate researchers to focus on developing robust scoring functions and to also fundamentally re-think frameworks for factuality guarantees beyond conformal filtering.
>
> **Other Models**: We want to clarify that our use of models that are modest in size was a deliberate choice for efficiency.
> 1. **Repetitive Task Cost**: Factuality claim scoring is a high-frequency, repetitive task. Using a large, computationally-heavy model for verifying every single claim would incur significant and recurring costs, making the system inefficient. Table 1 in the Appendix A.2.8 illustrates the FLOPs required for different models to generate texts. We can see that large models like DeepSeek-R1, even under the mixture of expert architecture, would still be 10 times more expensive than models of modest size.
> 2. **Democratization**: Using models that are of modest size provides a viable path for developers, researchers, and organizations without access to large-scale compute to build reliable, factually-grounded systems.
>
> That said, based on your suggestion, we ran additional experiments on the Qwen3-32B (and its 'Think' variant) and Qwen3-30B-A3B model in Section 4.3. We have updated the results in Figure 5. We observe that on the FActScore dataset, Qwen3-0.6B still performs the best across sufficient correctness and power. On the MATH dataset (Figure 27 in Appendix A.2.4), Qwen3-8B is competitive with its larger, 30B+ variants. These new results align with our previous finding that simply scaling the model size does not always help.
>
> We hope our response resolves your queries and we hope that you consider increasing the score. We are happy to answer any further questions.

---

### Author Response · Authors · 2025-11-16

We thank all the reviewers for their constructive feedback, insightful questions, and suggestions, which helped to clarify and strengthen the contributions of our paper.

The reviewers recognized several key strengths of our work:
- **Thorough Empirical Study** (tKcX, GfF5, fFdw): A systematic evaluation of conformal factuality for RAG-based LLMs across diverse tasks, multiple model families, and different scoring functions.
- **Novel and Critical Metrics** (tKcX, GfF5, hf6H): Non-empty Rate, Non-vacuous Empirical Factuality, Sufficient Correctness – that are essential for balancing factuality with informativeness, a key aspect that standard metrics fail to capture.
- **Analysis of Robustness and Failure Modes** (hF6H, G5F5, fFdw): A rigorous empirical study of the system's robustness to distribution shifts and distractor-filled contexts, which simulate real-world challenges.

We would like to take this opportunity to reiterate our core contribution:
1. To our knowledge, our work is the **first** to perform a comprehensive **systematic evaluation of conformal factuality for RAG-based LLM systems**—across diverse tasks, model families, and scoring functions.
2. We propose **novel metrics for evaluation** designed to better capture the “usefulness” of conformal factuality systems. Our metrics complement existing ones and provide evaluation beyond traditional empirical factuality, to address practical needs in real-world deployment.
3. Systematic **evaluation framework focusing on robustness to distribution shifts**, a key aspect for practical considerations. These contribute actionable insights to both practitioners and researchers.

We have addressed the reviewer's questions in detail in our individual responses. We look forward to addressing any further questions during the discussion period.

---

### Meta-Review · Area_Chair_QKDz · 2026-01-05

**Summary:**

This paper presents a systematic empirical study to integrate Conformal Factuality (CF) filtering with Retrieval-Augmented Generation (RAG), and introduces three new metrics to better capture the trade-off between factuality and usefulness. Reviewers acknowledge the thorough experimental evaluation across datasets and model families, and agree that the identified limitations in existing factuality metrics (e.g., empty or vacuous responses being treated as correct) are valid. However, there is broad agreement among reviewers that the paper’s contributions are largely incremental and empirical in nature. The core approach relies on a straightforward combination of established techniques (RAG and conformal prediction). While the proposed metrics are reasonable, reviewers question whether they provide insights beyond highlighting already-known trade-offs. And the concrete guidance on how practitioners should act based on the insights is unclear. Overall, while the study offers useful observations and careful analysis, the novelty and practical impact are not strong enough to meet the bar for acceptance. I therefore recommend rejecting the paper in its current form.

**Reviewer Concerns:**

The rebuttal clarified technical details, fixed presentation issues, added missing results (including larger models and metrics). But the key concerns remain regarding limited novelty beyond a systematic empirical study, unclear practical guidance on metric trade-offs and deployment, and insufficient justification of real-world impact.

**Reviewer Scores:**

All reviewers initially gave negative scores. Their ratings are unlikely to change significantly, as the key concerns regarding limited novelty, incremental contribution, and unclear practical impact remain unaddressed.

---

### Decision · Program_Chairs · 2026-01-26

Reject